# Single-cell chromatin accessibility profiling of acute myeloid leukemia reveals heterogeneous lineage composition upon therapy-resistance

Huihui Fan [1,2,12], Feng Wang[3,12], Andy Zeng[4,5], Alex Murison[4,5], Katarzyna Tomczak [3], Dapeng Hao [3], Fatima Zahra Jelloul [6], Bofei Wang[7], Praveen Barrodia[3], Shaoheng Liang[8,11], Ken Chen [8], Linghua Wang[3], Zhongming Zhao [2,9], Kunal Rai [3], Abhinav K. Jain [10], John Dick[4,5], Naval Daver[7], Andy Futreal [3] & Hussein A. Abbas [3,7 ✉]

Acute myeloid leukemia (AML) is a heterogeneous disease characterized by high rate of therapy resistance. Since the cell of origin can impact response to therapy, it is crucial to understand the lineage composition of AML cells at time of therapy resistance. Here we leverage single-cell chromatin accessibility profiling of 22 AML bone marrow aspirates from eight patients at time of therapy resistance and following subsequent therapy to characterize their lineage landscape. Our findings reveal a complex lineage architecture of therapy-resistant AML cells that are primed for stem and progenitor lineages and spanning quiescent, activated and late stem cell/progenitor states. Remarkably, therapy-resistant AML cells are also composed of cells primed for differentiated myeloid, erythroid and even lymphoid lineages. The heterogeneous lineage composition persists following subsequent therapy, with early progenitor-driven features marking unfavorable prognosis in The Cancer Genome Atlas AML cohort. Pseudotime analysis further confirms the vast degree of heterogeneity driven by the dynamic changes in chromatin accessibility. Our findings suggest that therapy-resistant AML cells are characterized not only by stem and progenitor states, but also by a continuum of differentiated cellular lineages. The heterogeneity in lineages likely contributes to their therapy resistance by harboring different degrees of lineage-specific susceptibilities to therapy.

[1] Department of Neurology, McGovern Medical School, The University of Texas Health Science Center at Houston, Houston, TX, USA. [2] Center for Precision Health, School of Biomedical Informatics, The University of Texas Health Science Center at Houston, Houston, TX, USA. [3] Department of Genomic Medicine, University of Texas MD Anderson Cancer Center, Houston, TX, USA. [4] Princess Margaret Cancer Center, University Health Network, Toronto, ON M5S 1A8, Canada. [5] Department of Molecular Genetics, University of Toronto, Toronto, Canada. [6] Department of Hematopathology, University of Texas M D Anderson Cancer Center, Houston, TX, USA. [7] Department of Leukemia, University of Texas MD Anderson Cancer Center, Houston, TX, USA. [8] Department of Bioinformatics and Computational Biology, The University of Texas MD Anderson Cancer Center, Houston, TX, USA. [9] Human Genetics Center, School of Public Health, The University of Texas Health Science Center at Houston, Houston, TX, USA. [10] Department of Epigenetics and Molecular Carcinogenesis, University of Texas MD Anderson Cancer Center, Houston, TX, USA. [11]Present address: Computational Biology Department, School of Computer Science, Carnegie Mellon University, Pittsburgh, PA, USA. [12]These authors contributed equally: Huihui Fan, Feng Wang. ✉email: habbas@mdanderson.org

Acute myeloid leukemia (AML) is a heterogeneous disease characterized by various morphologic, genomic, cytogenetic, and functional groups[1–3]. Morphologic heterogeneity of AML emerges from disrupted differentiation and is characterized by a spectrum of cells from CD34+ precursor cells to morphologically well-differentiated lineages[4,5]. With advents of molecular profiling and the identification of recurrent cytogenetic and genetic mutations, the morphology-based classification of putative cell lineages in AML was largely replaced by the defining recurrent molecular abnormalities[5]. Gene expression profiling garnered further insights into the biological and functional heterogeneity of AML cells[3,6–8]. More recently, single-cell RNA sequencing (scRNA-seq) revealed a more complex lineage hierarchal system supporting heterogeneity along the myeloid axis in newly diagnosed AML patients[9].

Unfortunately, 40–50% of AML patients have primary refractory disease or relapse shortly after remission (i.e., therapy-resistant)[10]. Since cellular lineages can impact resistance to therapy[11], it is critical to delineate the lineage composition of AML cells at the time of therapy resistance. Gene expression signatures revealed that therapy-resistant AML is characterized by enrichment for primitive hematopoietic stem cell (HSC) and progenitor characteristics[1,12]. Whether the AML primitive state at the time of therapy resistance represents a homogeneous cellular state or is composed of a continuum of different cellular lineages that impact resistance to therapy is not clear.

Since epigenetic changes precede the gene expression programs, it is critical to characterize the genome-wide chromatin accessibility to identify cellular lineages[13–19]. In support of this, hematopoiesis is indeed regulated by dynamic chromatin states and lineage-specific transcription factors (TFs) that dictate cellular fates[20–22]. The assay for transposase-accessible chromatin using sequencing (ATAC-seq) is a robust tool that assesses the epigenetic landscape via efficiently probing the chromatin accessibility of cells[23]. Akin to ATAC, the development of single-cell ATAC sequencing (scATAC-seq) uncovered new insights into regulatory chromatin states of individual cells that would otherwise be masked by bulk sequencing approach[20] and can be even more robust than scRNA-seq profiling[19,24]. Applying scATAC-seq in AML at time of therapy resistance can reveal cellular fates, and thus provides a higher resolution of intratumoral lineage compositions, and ultimately enabling a deeper understanding of AML heterogeneity that contribute to therapy resistance.

In this study, we applied scATAC-seq profiling on 22 bone marrow aspirates from eight therapy-resistant AML patients at time of therapy resistance and subsequent therapy to dissect the dynamic cellular states that constitute AML malignant cells. Since the cell of origin can impact responses to therapy, our goal was to delineate the lineage composition and epigenetic regulatory landscapes that likely contribute to therapy resistance in AML.

## Results

**Open-chromatin landscape in AML patients using scATAC-seq data**. We performed scATAC-seq profiling on 22 whole bone marrow aspirates collected at different treatment timepoints from 8 AML patients who received prior therapies and relapsed or were primary refractory to PD-1 blockade, hereafter referred to as therapy-resistant AML. All patients were treated on protocol[25] at the time of therapy resistance (pretreatment = timepoint A) with azacitidine and nivolumab (following treatment = timepoints B and C). Briefly, our cohort had a median age of 73 years (range 64–88) prior to receiving PD-1 blockade therapy. While on the treatment, 3 patients (PT1-3) were responders (R); 3 patients (PT4-6) were non-responders (NR) and 2 patients (PT7-8)

showed stable disease (SD) (Fig. 1a). Combining our published scRNA-seq gene expression data using the same cohort[7], we aim to reveal the chromatin landscape in these AML patients with a focus on malignant cells.

By applying Signac pipeline[26], a total of 59,321 mononuclear bone marrow individual cells passed quality control after excluding 2 low-quality samples, and were grouped into 2 broader clusters, i.e., AML malignant cells and normal cells in the tumor microenvironment (TME) cells (Fig. 1b), with granular cell labels shown in Fig. 1c. Open-chromatin variations measured using standard deviations in peak signals per sample were shown for both AML malignant and TME cells (Supplementary Fig. 1a). Two extra healthy controls were included for this comparison. As demonstrated, normal samples NL1 and NL2 were very similar in terms of the distributions for open-chromatin variations. Similarly, the variation distributions across different timepoints for TME cells in each patient did not differ much from each other. However, variation distributions within malignant cells tended to show larger shift across different timepoints which also corresponded to their clinical outcomes. For example, open-chromatin variations shrunk at timepoint C (closer to time of lack of response to azacitidine/nivolumab therapy) while comparing to timepoint A (prior to therapy initiation) in treatment-responsive patients PT2 and PT3.

To yield a robust classification between malignant and TME cells, we utilized SC3 consensus clustering algorithm[27] to assess the stability of two broader clusters using different subsets of peaks (Supplementary Fig. 1b–d). Cluster stability continued to improve from using all peak set, to promoter-marked, and to enhancer-marked peaks (Supplementary Fig. 1e). In particular, distal enhancer-marked peak set showed the best performance for malignant cell classification. Broader cluster labels were thus inherited from the classifications using peak set derived from distal enhancers. Annotation of the scATAC-seq clusters through integration with our previously published scRNA-seq profiles of the same bone marrow aspirates[7] uncovered 12 major bone marrow cell types of erythroid, lymphoid, and myeloid lineages (Fig. 1d). The number of peaks present in each cell type varied significantly, with malignant cells showing the most and plasma cells showing the least number of peaks (Supplementary Fig. 1f). Genomic annotations for sample-wise peak sets demonstrated similar distributions, with promoter-marked peak proportions differed more than the rest of the genomic locations, such as distal intergenic regions (Supplementary Fig. 1g).

Marker peak analysis revealed distinct peak accessibility across diverse TME cell types and malignant cells (Fig. 1e), consistent with previously observed cell-type-specific chromatin accessibility profiles[15,28–30]. To further link marker peaks with marker genes, we integrated our previously published scRNA-seq profiles and calculated the top 10 differentially expressed genes (DEGs) within each putative cell type (Fig. 1e). For instance, CD79B[31] was one of the top marker genes linked to B-cell-specific peak set (K-means cluster 13), whereas the Tetraspanin CD82[32] was ranked as one of the top markers in late erythroid cell (K-means cluster 12). However, as shared among various cell types of late erythroid, CD4, CD8, and NK, cluster 8 marker peak set did not exhibit any distinct peak-gene annotations.

By re-applying ArchR pipeline, we were able to verify that more than 90% of the malignant cells called by Signac were consistent with archR predictions (Supplementary Fig. 2a). Cell label annotations of the scATAC-seq clusters by archR uncovered 9 bone marrow cell types of erythroid, lymphoid, and myeloid lineages (Supplementary Fig. 2b). Marker peak analysis revealed distinct peak accessibility across the healthy and leukemic cellular subtypes (Supplementary Fig. 2c), consistent with cell-type-specific chromatin accessibility profiles[15,28–30]. To further

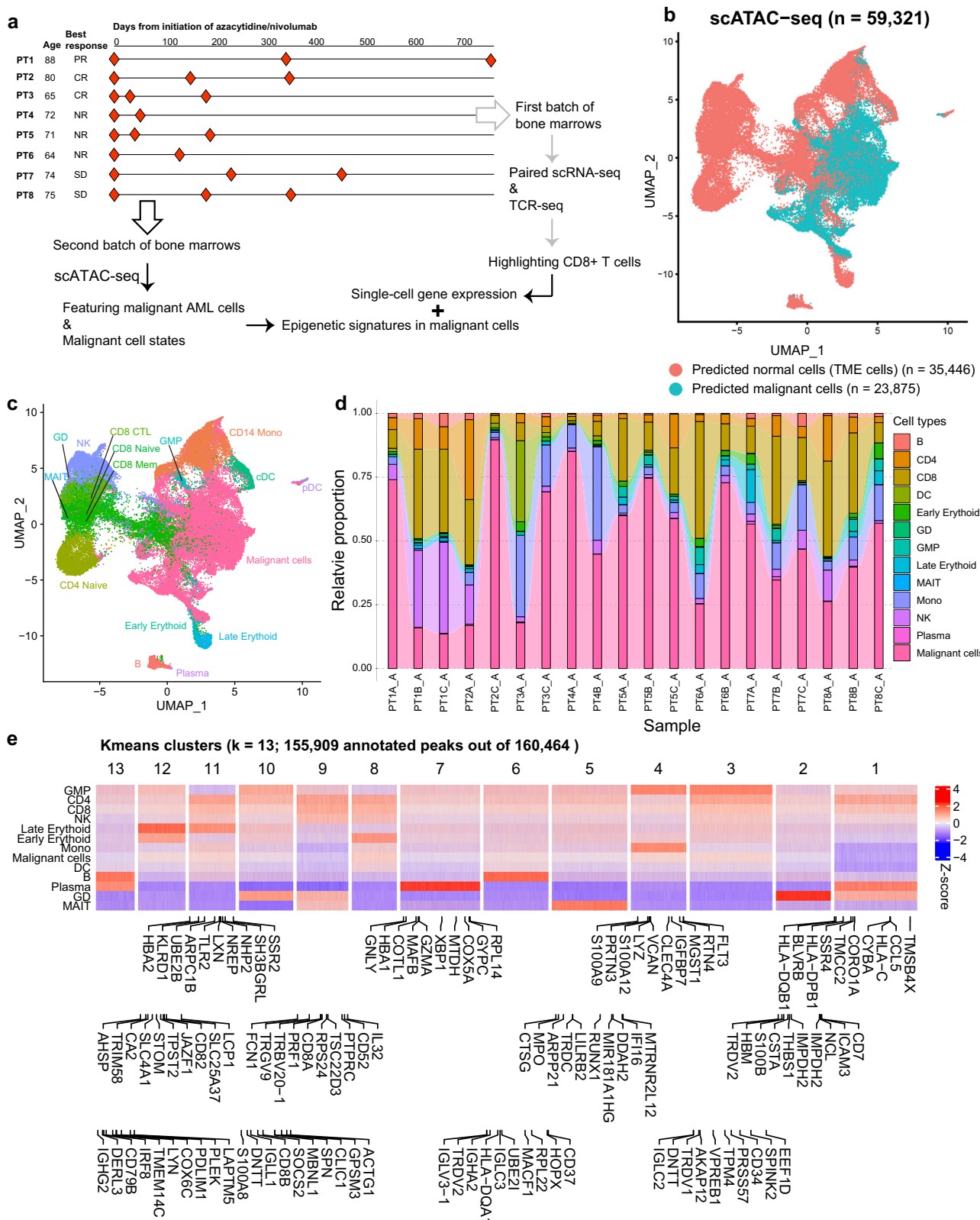

validate the cluster annotation, we visually inspected the tracks of canonical gene markers (Supplementary Fig. 2d) and calculated the lineage-defining gene scores, which represent the overall chromatin accessibility at the gene body and promoter regions[29,33–35]. For instance, T cells had the highest mapped gene expression by integrating scRNA-seq data for CD3E and CD8B, whereas erythroid cells had high gene score for HBB

inferred with their epigenetic accessibility (Supplementary Fig. 2d, upper panel), that were consistent with their predicted gene activity scores using scATAC-seq data in the putative cell types (Supplementary Fig. 2d, lower panel). Next, we analyzed the differential accessible chromatin regions for enriched DNA-binding motifs of lineage-specific transcription factors, which correlate with cell identity[22]. Our accessibility profile of

**Fig. 1 Open-chromatin landscape in AML patients profiled using scATAC-seq. a** An introduction of the recruited cohort for our study which included eight patients. Longitudinal bone marrow samples were extracted per patient for sequencing. First batch of samples were profiled using paired scRNA-seq and TCR-seq, with a highlight of CD8 T cells (published), while second batch of samples were profiled using scATAC-seq and centered on the analyses of malignant cells by including gene expression as a complement. **b** UMAP of scATAC-seq clusters colored by tumor microenvironment (TME) and malignant cells. **c** UMAP of scATAC-seq clusters colored and labeled by granular cell types. **d** Barchart showing relative proportions of predicted cell types across different samples. **e** Heatmap showing all annotated peaks ($n = 155,909$) aggregated by cell types. Each column represents a peak region, while each row represents a cell type. Peaks are split into 13 K-means clusters, with top ten differentially expressed genes (calculated using our published scRNA-seq data) labeled along each cluster.

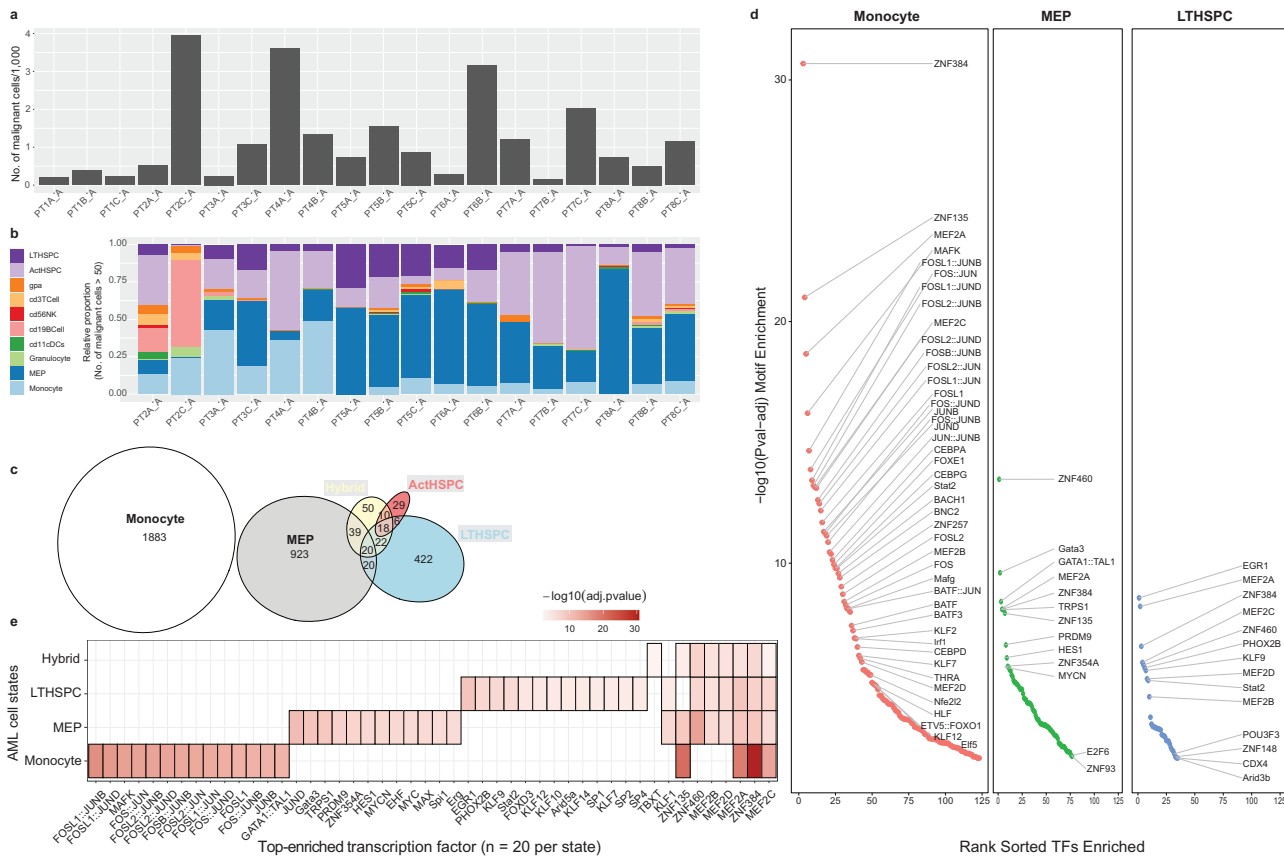

**Fig. 2 Diverse cell states in therapy-resistant AML malignant cells. a** Barchart showing the number of malignant cells across different samples. **b** Relative proportion of malignant cells resembling each normal cell type. Only samples with more than 50 malignant cells are shown. **c** Venn diagram showing the number of differentially accessible regions (DARs) while comparing malignant cells with cell states of monocyte, MEP, Act (active) HSPC, LT (long-term) HSPC and hybrid to malignant cells without any cell state assigned. Hybrid cell state means more than one cell state is assigned to one cell. **d** Enrichment analysis of transcription factors (TFs) within each AML cell state. TFs are ranked based on their enrichment significance level on x axis. Arrows mark functional terms discussed in the main text. **e** Heatmap showing top 20 enriched TFs per cell state, colored by enrichment significance, with columns labeled as TFs and rows labeled as cell states.

transcription factor enrichment scores confirmed canonical transcription factor regulators of the identified cell types (Supplementary Fig. 2e).

**Hematopoietic progenitor-like and stem cell-like AML cell states are dominated in therapy-resistant AML patients.** Next, we focused on AML malignant cells to identify diverse cell states involved in our study cohort. Using label transfer from scRNA-seq data, AML malignant cells were predicted using distal enhancer-based peak set in scATAC-seq data (Fig. 2a). The number of malignant cells varied greatly across different samples, ranging from dozens to a couple thousands. This could be partially due to the uniqueness of patients with different responsive states across different sampling timepoints. We then leveraged accessibility signatures which was based on sorted well-defined hematopoietic cells from different lineages[36] to refine cell states in

AML cells. By assigning an accessibility Z-score using chromVAR[37] for ten chromatin accessibility signatures spanning normal hematopoiesis to more differentiated cells, as previously described[36], we assigned each single cell a putative identity based upon the signature which showed the highest positive enrichment. Samples with less than 50 malignant cells were excluded from cell state proportion calculations (Fig. 2b and Supplementary Fig. 3). Our results suggested that populations of AML cells in each sample showed highest enrichment of early stem cell and progenitor signatures but was also constituted of signatures of more differentiated cells which included erythroid, myeloid and lymphoid lineages. We found different degrees of stemness including gradients of long-term (LT) (i.e., quiescent) and activated (ACT) hematopoietic/progenitor stem cell (HSC/HSPC) priming within each patient, and across different patients (Supplementary Fig. 4a). Of note, a considerable proportion of AML

cells had chromatin accessibility primed for erythroid (erythroid progenitors) and myeloid-erythroid progenitors (Supplementary Fig. 4b). Monocyte-like AML malignant cells were more present in responsive patients (PT2 and PT3) when comparing to non-responsive patients and patients with stable disease (from PT5 to PT8) (Supplementary Fig. 4c). On the contrary, megakaryocyte erythroid progenitors (MEP)-like and stem cell-like (LTHSPC and ActHSPC) AML cells were significantly dominated in patients with non-responsive and stable disease (Supplementary Fig. 4a, b). Compositions of B-cell-like AML cells were not significantly altered across different patient groups (Supplementary Fig. 4d). This actually supported that therapy-resistant AML cells are enriched for stem-like and progenitor lineages[1,38]. We next focused on patient 3 who acquired a cytogenetic deletion in chromosome 7 at timepoint C; patient 4 who had an evolution of CD34- CD14+ monocytic cells at the time of disease resistance (timepoint B) and patient 7 who had stable disease for >1 year while on therapy (timepoints B and C). Patient 3 and patient 7 showed increasing primitive stem cell signatures/genes along the timepoints while patient 4 showed increasing monocyte signature/genes and decreasing primitive stem cell signatures/genes (Fig. 2b). Importantly, these hierarchal distributions also persisted following treatment (timepoints B and C), although the proportion of each signature varied across samples. These observations were consistent with their clinical manifestations.

**Single-cell RNA profiling of AML cells validated the enrichment of early progenitor cell states**. We next leveraged the scRNA-seq dataset by Granja et al. that harbored >30,000 cells highly enriched for all stages of differentiation, including stem and progenitor cells[29]. When projecting our previously published single-cell transcriptional profile of AML malignant cells onto the above normal hemopoietic reference using Symphony[39], we found most of the AML cells were mapped to HSC or early progenitors (Supplementary Fig. 5a). However, we also found cells projected onto differentiated cells, such as monocytes. As expected, AML malignant cells largely resembled normal hematopoietic and early progenitors albeit at slightly different proportions within each patient (Supplementary Fig. 5b) when comparing to our scATAC-seq-based cell states annotations (Fig. 2b). Our findings support distinct lineage priming patterns across cells using both scRNA-seq and scATAC-seq data, with scATAC-seq profiling showing much higher resolution in lineages (Fig. 2b).

**Malignant cell states-specific transcriptional regulatory landscape**. To characterize the specific molecular underpinnings of each AML malignant cell state, we then calculated differentially accessible regions (DARs) by comparing AML malignant cells assigned with unique cell states to AML cells without any cell states assigned (Fig. 2c). As indicated, MEP-like and HSPC-like AML cells share some DARs, while majority of DARs remained unique to each cell state. Monocyte-like AML cells did not share any DARs with hybrid-, progenitor-, and HSPC-like AML cells. To delineate the higher level of regulations within each AML cell state, transcription factor (TF) enrichment analysis was then carried out (Fig. 2d). We observed an extensive and almost exclusive enrichment of AP-1 regulation was observed in monocyte-like AML cells (Fig. 2d, e). The activator protein-1 (AP-1) is a collection of TFs that included four sub-families: the Jun (c-Jun, JunB, JunD), Fos (c-Fos, FosB, Fra1, Fra2), Maf (c-Maf, MafB, MafA. Mafg/f/k, Nrl), and the ATF-activating TFs (ATF2, LRF1/ATF3, BATF, JDP1, JDP2)[40], characterized by pleiotropic effects and a central role in the immune system such as T-cell activation, and T-cell anergy and exhaustion[41,42]. AP-1

proteins regulate immunomodulatory processes, cell proliferation, apoptosis and growth[43], that are also implicated in the pathogenesis of leukemia and lymphoma where these TFs can act as oncogenes[44,45]. Myocyte enhancer factor 2 (MEF2) are a group of proteins consisting of four distinct members MEF2A, B, C, and D. Mef2 has a wide variety of functions in different cells including cardiomyocytes and HSCs. Particularly, Mef2c in comparison to the other family members is preliminarily expressed and involved in the mouse hematopoiesis[46]. It is differentially expressed in the progenitor cells and regulates hematopoietic development. Previous research has reported that Mef2c is abundantly expressed in the HSCs and common lymphoid progenitor cells (CLPs). Whereas the expression declines when common myeloid progenitor cells (CMPs) differentiate into much committed forms like granulocyte myeloid progenitors and MEPs. We observed TF MEF2A and MEF2C were shared across different AML cell states, while MEF2B and MEF2D were missing from monocyte-like AML cells (Fig. 2e). We then aimed to explore the cell state-unique gene signatures and their driven biological functions in therapy-resistant AML malignant cells.

**Unique gene signatures driven by the diverse AML cell states-associated chromatin accessibility alterations**. To further characterize the diverse AML cell states, we integrated scATAC-seq and scRNA-seq data to link cell states-associated DARs with DEGs by comparing malignant cell states with their normal counterparts (Supplementary Fig. 6 and Supplementary Data 1). As shown, monocyte- and MEP-like AML cell states in scATAC-seq space resembled the monocyte- and early erythroid-like AML cell states in scRNA-seq space, respectively. In particular, open DARs corresponded with upregulated DEGs in the same or similar AML cell states across both modalities. Compared to monocyte- and MEP-like AML cell states, HSPC-like AML cells showed fewer differential events in chromatin accessibility. We also observed that chromatin accessibility tended to be more open when comparing malignant lineages with their normal counterparts (Supplementary Fig. 6). Differential events were then binarized to retain information for differential directions (Fig. 3a). DAR-DEG pairs were kept only if both showing a significant differential regulation in the same direction, for instance open DAR and upregulated DEG, or closed DAR and downregulated DEG. Cell states-associated gene signatures were thus defined using these DARs-linked DEGs with the same differential directions (Fig. 3b). In total, we identified 324, 117, 22, and 5 unique cell state-related features in monocyte-, MEP-, LTHSPC-like and ActHSPC-like AML cells.

Using cell states-associated gene signatures, we performed a functional enrichment analysis using functional gene sets from the Molecular Signatures Database (MSigDB)[47] (Supplementary Data 2). Hierarchical tree plots were constructed based on the pairwise similarities of the enriched functional terms using Jaccard's similarity index (Fig. 3c–e). As demonstrated, Gene set (EPPERT HSC R)[48] containing genes upregulated in HSC-enriched populations compared to committed progenitors and mature cells was significantly enriched in our LTHSPC-associated gene signature (Fig. 3c). Similar functional terms also included "HAY BONE MARROW CD34 POS HSC"[49]. Collectively, HSPC-like AML cell state was confirmed and characterized by the significant overlap with previously curated AML signature gene sets (VALK AML CLUSTER 1 and VALK AML CLUSTER 15)[50], the activation of PI3K-Akt signaling pathway (REACTOME PI3K-AKT SIGNALING IN CANCER)[51], the enhanced metastasis and mobility by MET signaling pathway (REACTOME SIGNALING BY MET)[52], the dysregulated cell–cell adhesion (GOBP REGULATION OF CELL CELL ADHESION and GOBP

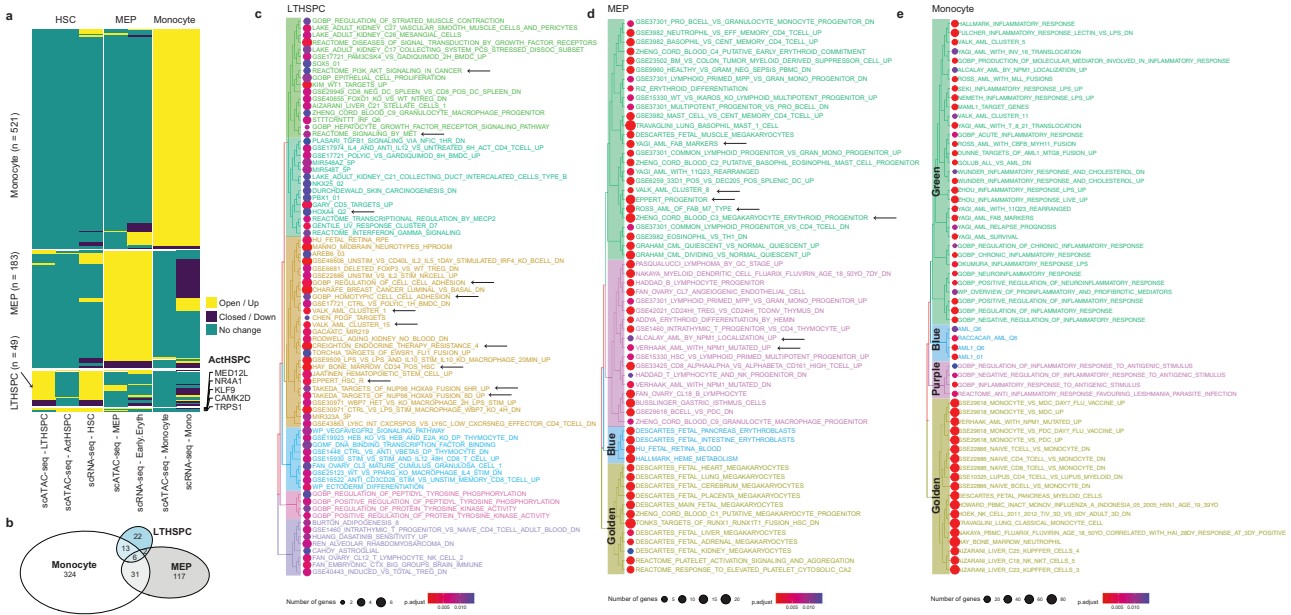

**Fig. 3 AML cell state-specific chromatin accessibility-driven gene signatures and their functions. a** Heatmap showing DARs and their linked DEGs, with five DAR-linked DEGs in active HSPC (ActHSPC) shown along the heatmap. Fold changes are visualized as binarized values, with open (>0) and closed (<0) for DARs, and up (>0) and down (<0) for DEGs. Rows are split based on AML cell state-specific DARs. **b** Venn diagram showing overlaps among the three gene signature sets, with ActHSPC signature gene set excluded. Functional enrichment treeplot for the three AML cell state-specific gene signature sets are shown in (**c–e**), respectively.

HOMOTYPIC CELL CELL ADHESION), the increased therapy resistance (CREIGHTON ENDOCRINE THERAPY RESISTANCE 4)[53] and a putative poor survival marked by the enrichment of TF HOXA4 target genes (HOXA4 Q2)[54]. The enrichment of the other homeotic TF HOXA9-related gene features suggested its inhibition could be an interesting strategy against AML[55]. As a leukemic driver gene, *HOXA9* in AML is associated with cell proliferation, differentiation blockade, increased malignancy, and self-renewal maintenance in progenitor cells[56]. Aside from the enrichment of AML-specific[50] and progenitor-related[57] functional terms, MEP-like AML cells were also enriched in FAB subtype-associated markers from pediatric AML patients[58], as well as AML patients with *NPM1* mutations[59,60] (Fig. 3d). Interestingly enough, we observed MEP signatures from different fetal organs, such as heart, lung, and liver, were significantly enriched in our MEP-like AML gene set (Fig. 3d, golden and blue branches). Similar observation was demonstrated with monocyte-like AML cells, that gene signatures from monocytes regardless of their origin organs highly resembled the gene features we extracted from monocyte-like AML cells (Fig. 3e, golden branch). AML-specific functional terms (Fig. 3e, blue branch) and inflammation-dominated functional terms (Fig. 3e, green and purple branches) were excessively observed in monocyte-like AML cells.

**Pseudotemporal analysis reveals chromatin accessibility of canonical transcription factor regulations.** We next leveraged the longitudinal inferred trajectory with supervised pseudotime ordering to understand the AML longitudinal topology and identified variability in lineage trajectories following treatment (Fig. 4a). Starting from HSPC cells, six different lineages were thus identified (Fig. 4b). We extracted AML cells underlying each trajectory and overlaid them with the diverse AML cell states (Fig. 4c and Supplementary Fig. 3). As expected, progenitor-like HSPC- and MEP-like AML cells were shared mostly across different lineages, while monocyte- and B-cell-like AML cells tended

to aggregate in certain trajectory branches, such as Y_74 and Y_155. We also performed hematopoietic accessibility signatures analysis per trajectory and overlaid the trajectory-related peak set with genomic features (Fig. 4d). As shown, the underlying peak set from trajectory Y_74 with dominated monocyte-like AML cells was highly enriched in exons (i.e., 1st exons and other exons) and 3' UTR regions. We then calculated TF binding enrichment using trajectory-related peak sets (Fig. 4e) and ranked the uniquely enriched TFs per trajectory by combining the top 30 most enriched TFs across all the trajectories (Fig. 4f). As illustrated, transcription factor encoded by gene *SPI1* is specifically expressed in myeloid cells and B-lymphocytes[61], which was among the top-ranked TFs and also shared across all six lineages (Fig. 4e, red arrows). As expected, Spi1 was ranked as the top one in trajectory Y_155 that was dominated with B-cell-like AML cells (Fig. 4c). We also extracted top-enriched TFs that were unique to each trajectory (Fig. 4f). We observed the uniqueness of the trajectory-based regulatory landscape, with lineage Y_155 and Y_209 similar to each other in regulating immune responses[62]. Noticeably, several GATA family TFs, especially GATA2, were among the top-unique TFs enriched in trajectory Y_17, indicating a potential adverse prognosis in AML patients driven by this lineage[63]. Based on the annotations of top-enriched unique regulators, we predicted that trajectory Y_161 was involved in tumor immunity and drug response[64]; Y_107 attenuated apoptosis through induction of autophagy in AML cells[65]; and Y_74 could be linked to cancer metabolism, such as enhanced mitochondrial output and energy production[66,67]. The diverse and unique functions driven by different trajectories, further expanded our understanding of the vast amount of heterogeneity within AML. These findings also suggest that dynamic changes in the transcriptional regulatory landscape marked by epigenetic accessibility following treatment could modulate cellular states.

We next leveraged the longitudinal inferred trajectory with supervised pseudotime ordering to understand AML longitudinal topology and identified variability in lineage trajectories following

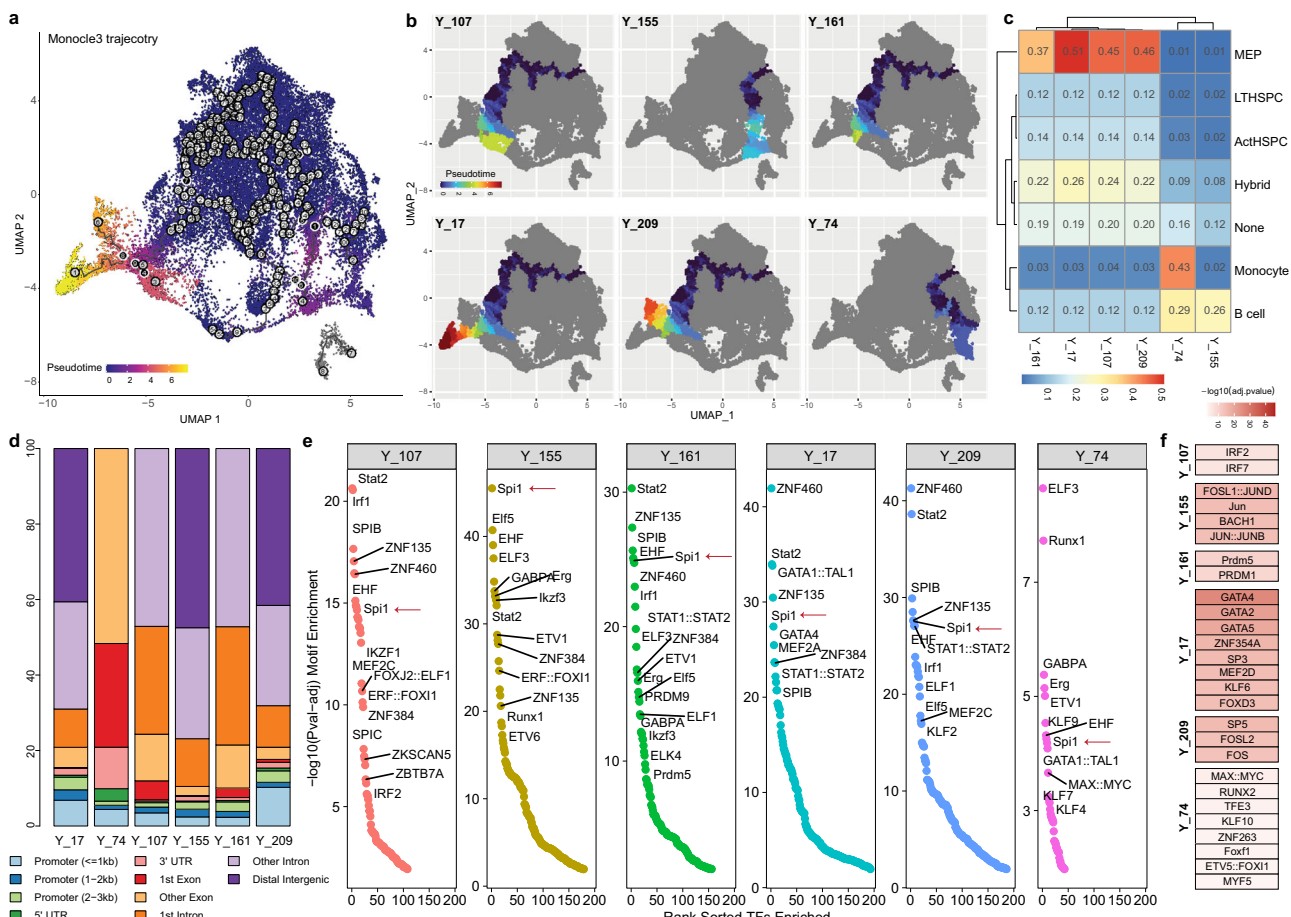

**Fig. 4 Monocle3 trajectory inference using AML malignant cells and trajectory-unique regulatory landscape. a** UMAP showing the trajectories across all eight patients, colored by inferred pseudotime. **b** UMAP showing six distinct cellular trajectories. **c** Heatmap showing the percentage of each AML cell state mapped to each trajectory. **d** Genomic distributions of trajectory-related open-chromatin regions. **e** Regulatory landscape across six distinct trajectories. **f** Top-unique transcription factors enriched for each trajectory.

treatment (Supplementary Fig. 7a). Trajectory analysis revealed dynamic changes of accessibility of genes associated with stemness (CD34, MLLT3, PROM1, GATA2, FLI1)[68] as well as those of lineage commitment related genes such as CD14, MPO, ALAS2, MPEG1, and CD19 (Supplementary Fig. 7b). We also performed hematopoietic accessibility signatures analysis and overlayed the enrichment score along the trajectory (Supplementary Fig. 7c). We focused our analysis on patient 3 who acquired a cytogenetic deletion in chromosome 7 at timepoint C; patient 4 who had an evolution of CD34- CD14+ monocytic cells at time of disease resistance (timepoint B) and patient 7 who had stable disease for >1 year while on therapy (timepoints B and C). Patient 3 and patient 7 showed increasing primitive stem cell signatures/ genes along the trajectory while patient 4 showed increasing monocyte signature/genes and decreasing primitive stem cell signatures/genes (Supplementary Fig. 7b, c). Motif enrichment along trajectory also showed consistent results. Specifically, Patient 3 showed transitions from differentiation-related motifs (such as CEBP family) to progenitor-related motifs (such as GATA family) (Supplementary Fig. 7d), while patient 4 showed an opposite pattern (transitioned from GATA enriched to CEBP and SPI1 enriched) (Supplementary Fig. 7e). Patient 7 also showed motif enrichment of LMO2, which is an essential transcriptional factor for primitive hematopoiesis, at the late stage along the trajectory (Supplementary Fig. 7f). These findings suggest dynamic changes in epigenetic accessibility following treatment that could modulate cellular states.

**Chromatin accessibility reveals the continuum of stem, progenitor, and lineage-restricted priming**. To further characterize the cell states in therapy-resistant AML cells, we calculated DARs by comparing AML cells with cell states assigned to their normal counterparts. By doing so, we generated 47, 74, 166, and 2101 DARs for monocyte-, HSPC-, MEP- and B-cell-like AML cells separately (Supplementary Data 3). Each cell state-related DARs set was treated as a set of meta-module signatures, which was later applied to score all the AML malignant cells. A cellular hierarchy was constructed to demonstrate the continuum of the diverse AML cell states (Fig. 5a). As indicated, stem, progenitor, and lineage-restricted monocyte-like AML cells were mostly parsed into three quadrants, while B-cell-like AML cells were less prevalent comparing to the other three AML cell states. By excluding feature sets with DARs less than 100, we computed and ranked their significantly enriched TFs (Fig. 5b). As expected, GATA family transcriptional factors, such as GATA2 and GATA3, were uniquely enriched in MEP-like AML cells. GATA2 and GATA3 play essential roles in the development and maintenance of hematopoietic systems. In particular, GATA2 is crucial for the proliferation and survival of early hematopoietic cells and is also involved in lineage-specific transcriptional regulation[69]. The disrupted biological function of GATAs in various hematologic disorders are emerging, especially in AML[69]. Consistently, we observed similar regulatory programs in trajectory Y_17 in which more than 50% of the MEP-like AML cells were mapped (Fig. 4c, f). Kruppel-like transcription factors (KLFs) were shown

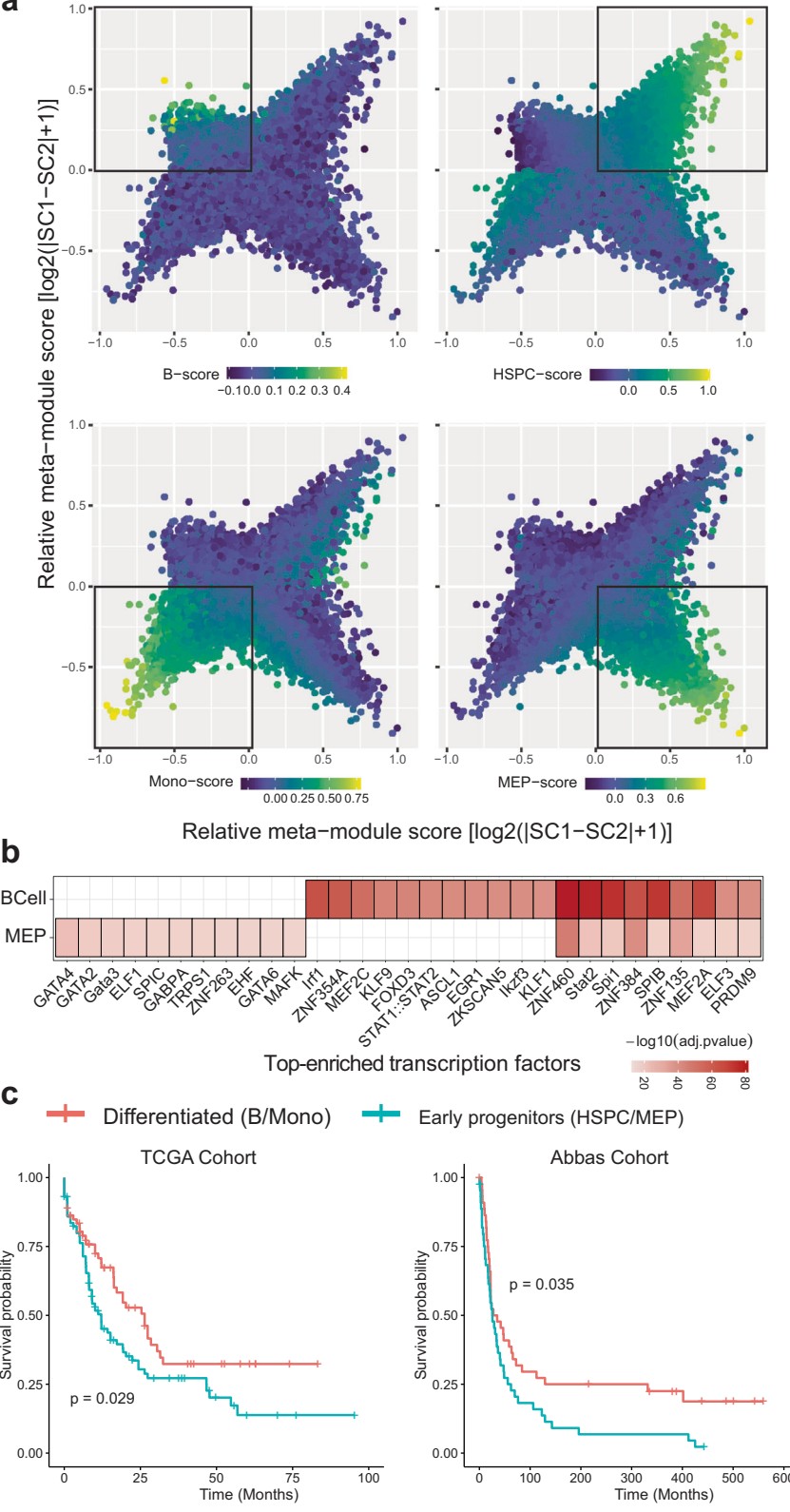

**Fig. 5 Malignant cell state-specific features and their survival relevance. a** The continuum of AML cell state scores defined by differential chromatin accessibility via comparing to their normal counterparts. **b** Heatmap showing top 20 most enriched transcription factors for AML cell states of B cell and MEP. **c** Survival analysis using the Cancer Genome Atlas (TCGA) AML cohort. Patients commonly high in B cell- and monocytes-derived scores are combined to generate the differentiated group, while patients commonly high in HSPC- and MEP-derived scores are combined to yield early progenitor group. Independent validation on prognostic values of our cell state-specific features is shown on the right using Abbas Cohort.

to be among the top-enriched TFs in B-cell-like AML cells. Specifically, gene *KLF9* can be applied as a prognostic marker in combination with other three markers ENPP4, TUBA4A, and CD247 to predict the overall survival of AML patients[70].

Lastly, we extracted DARs-lined gene features from HSPC-, MEP-, B-cell-, and monocyte-like AML cells, respectively. By linking DARs to genes, we extracted four sets of gene signatures that represented two broader lineages of early progenitors (HSPC and MEP) and differentiated cells (monocyte and B cell). Using The Cancer Genome Atlas (TCGA) AML cohort, each patient was scored using the above four sets of gene signatures[71], before collapsing into two groups (see "Methods"). We concluded that early progenitor-marked AML patients exhibited a less favorable survival prognosis when comparing to differentiated cells (Fig. 5c). This was also partially in line with our previous observations that AML malignant cell states that resembled differentiated cell types were more prevalent in some responsive AML patients (Fig. 2b), for example, B-cell-like AML cells were dominated in responsive patient PT2 at timepoint C.

## Discussion

AML is a devastating disease characterized by differentiation arrest[72] and is associated with high rate of therapy resistance, ultimately translating into short survival[73]. Our knowledge of AML lineage classification has been largely based on newly diagnostic samples, while less is known about AML composition at time of therapy resistance. It is critical to understand the lineage composition of AML cells at time of therapy resistance as cell lineages impact responses to therapy. Since lineage priming in the hematopoietic cells occurs largely at the epigenetic level and precedes transcriptional changes[68,74], we leveraged single-cell chromatin accessibility to understand the lineage composition of therapy-resistant AML cells. Additionally, our in-house generated single-cell chromatin accessibility data can be served as a rich resource for large-scale data integration and mining to the AML research community.

Our findings revealed intratumoral lineage heterogeneity characterized by AML cells harboring chromatin accessibility programs spanning the hematopoietic lineage continuum. Specifically, at time of therapy resistance, AML cells were enriched for hematopoietic stem cells and progenitors, in agreement with previous gene expression-based studies[12,75,76]. Among the stem and progenitor cell states, there was a spectrum between long-term and activated hematopoietic stem cell states that can contribute to relapse. However, our chromatin accessibility profiling revealed that AML cells were also primed for erythroid, myeloid, and lymphoid cells which includes early progenitors and cells at different stages of the differentiation spectrum. These components are similar to those seen in normal hematopoiesis. The multilineage state at time of therapy resistance likely contributes to the heightened resistance of AML cells. Importantly, subsequent treatment did not deter the heterogenous lineage composition of AML cells. Rather, AML cells maintained their stem cell-like epigenetic landscape, while propagating the hierarchal diversity in the lineages, although the relative abundance of some of those lineages changed with time. Our patient-derived longitudinally inferred trajectory with supervised pseudotime ordering revealed the transition of cellular states based on the dynamic changes of chromatin accessibility following treatment. Since the cell of origin can impact the response to therapy[11], we propose that the acquisition of a differentiation spectrum not only contributes to the generation and maintenance of leukemic stem cells as previously suggested, but it may also create distinct barriers for therapeutic responses ultimately leading to resistance and persistence of AML.

Noteworthy, early progenitor-like AML cell states-driven gene signatures are less favorable in survival prognosis using The Cancer Genome Atlas AML cohort, when comparing to the more differentiated AML cell states-driven gene signatures. Using previously published treatment-naive AML cohort[6], we validated the prognostic values of our cell states-derived features. Specifically, we have observed a discrepancy in regulatory potential between open-chromatin-marked signature regions and their linked genes. Similar phenomena regarding the low correlations between open-chromatin signals and their linked gene expression profiles have been studied using rat adipose and muscle tissues, with the highest correlation nearing 0.4 within promoter regions in muscle[77]. ATAC-seq enables the genome-wide identification of transcription factor binding events to orchestrate gene expression[14]. Since gene regulation is a complex process which involves the interplay of multiple layers of genomic and epigenomic regulations[78], it is especially difficult to reveal the complete picture for gene regulation when scRNA-seq and scATAC-seq data are not profiled using the exact same single cells, with one layer of epigenetic mark, i.e., chromatin accessibility. In addition, technical challenges are manifested in cross-referencing malignant AML cell states with normal hematopoiesis references. The ultimate workaround is to apply multi-omics profiling that outputs RNA expression and open-chromatin map simultaneously for each cell to characterize how open-chromatin states precede the changes in transcriptomics.

The human hematopoietic system is a hierarchal system which is tightly regulated by dynamic changes in chromatin accessibility[20,21] and transcription factors that dictate cell lineages[22] Similar to normal hematopoiesis, the rewiring of cellular transitions in therapy-resistant AML malignant cells was afforded by dynamic dependency on canonical transcriptional regulators of lineage transitions such as GATA1, RUNX, and POU2F2[79], and regulators of cellular transformation such as transcription factor AP-1[80]. AP-1 is composed of 18 dimeric complexes, that are critical in a wide range of cellular processes, including inflammation, proliferation, differentiation, and apoptosis, especially during oncogenesis[81,82]. In line with its functional profile, a nearly exclusive and unique enrichment of AP-1 was observed in monocyte-like AML malignant cells. Our data also revealed significant enrichment for HOXA family-related functions in stem-like AML malignant cells that is consistent with their role as master transcriptional regulators of lineage development and leukemogenesis[83–85]. Specifically, HOXA clusters have been implicated in *MLL*-rearranged and *NPM1* mutated AML inducing a stem cell phenotype[83–89]. Altogether, our findings suggest that AML cells exploit the normal transcriptional machinery, which would create an additional barrier in targeting these programs without incurring damage on normal hematopoiesis programs.

Our findings demonstrate that therapy-resistant AML cells are indeed characterized by primitive and stemness states, but also composed of other cellular lineages revealing significant intratumoral lineage heterogeneity. Further, AML cells exploit epigenetic regulators that are also shared with normal hematopoiesis, ultimately recapitulating a hematopoietic hierarchy. Therefore, this multilayered leukemia priming states creates a major barrier in targeting the complex architecture and provides AML cells with multiple functional layers of resistance. Not surprisingly, therapies that induced epigenetic-based differentiation of AML into homogeneous lineage states, such as IDH1, IDH2 and menin inhibitors[90–93] have demonstrated promising therapeutic options in AML. Since AML cell lineages are tightly regulated by epigenetic reprogramming, implementing epigenetic remodeling drugs that may potentially drive AML into a single-layered differentiated state that may overcome resistance.

## Methods

**Human subjects and treatment regimen.** Bone marrows from eight adult (>=18 years of age) patients on NCT02397720 protocol were included in this study. All patients had histologically proved relapsed or refractory acute myeloid leukemia. Timepoint A was collected at the time of proven relapse or primary refractoriness. Timepoints B and C were collected during the treatment with azacitidine 75 mg/m$^2$ (administered intravenously on days 1–7 over 60–90 min or subcutaneously), and nivolumab 3 mg/kg (administered as a 60–90 min intravenous infusion) on days 1 and 14 of each cycle treated on NCT02397720 protocol. Written informed consent was obtained from all participants. The study was conducted in accordance with the Declaration of Helsinki and had IRB approval.

**Bone marrow cell preparation.** All bone marrow samples were stored in liquid nitrogen and retrieved right before sample processing. To maximize the cellular viability recovery, samples were processed in batches according to in-house developed protocol and 10x Genomics "Demonstrated Protocol Cell Preparation Guide" (Document CG00053). Briefly, cells were gently thawed in water bath at 37 °C until are partially thawed and immediately placed on ice. Next, cells were gently transferred to a 10 ml media (10 ml alphaMEM + 20%FCS) and centrifuged (1500 rpm for 5 min). After removal of the supernatant, the cell pellet was carefully resuspended in 10 ml enriched media (alphaMEM+20%FCS supplemented with 500 µL Heparine, 15 µL DNase and 500 µL MgSO$_4$), followed by incubation in 37 °C for 15 min. After incubation, cells were centrifuged and gently washed twice in 1.5–3 mL of 0.04% BSA in PBS. Additionally, cells were passed through strainer (0.35–0.4 µm Flowmi Cell Strainer) to eliminate cell clumps. Next, cells were stained with 0.4% Trypan blue and quantified and assessed for viability using the cell automated counting machine Cellometer Mini (Nexcelom, Lawrence, MA, USA), as well as using standard hemocytometer and light microscopy.

**Nuclei isolation for single-cell profiling.** The nuclei isolation was performed according to the 10x Genomics "Demonstrated Protocol Nuclei Isolation ATAC Sequencing RevD" (CG000169) with some in-house optimized modifications. Briefly, cells were thawed and prepared as indicated in the previous section counted and maintained on ice. Cell suspension of ~0.9–1 million cells were centrifuged (300 rcf for 5 min at 4 °C) followed by gentle removal of the supernatant without disrupting the cell pellet. Next, cells were lysed in 100 µl of chilled Lysis Buffer (10 mM Tris-HCl (pH 7.4); 10 mM NaCl; 3 mM MgCl$_2$; 0.1% Tween-20; 0.1% Nonidet P40 Substitute; 0.01%; Digitonin and 1% BSA) with incubation on ice set for 8 min (as optimized). Next, cells were immediately washed by adding 1 mL of chilled Wash buffer (10 mM Tris-HCl (pH 7.4); 10 mM NaCl; 3 MgCl$_2$; 0.1% Tween-20 and 1% BSA) and centrifugated (500 rcf for 10 min at 4 °C). The resulting nuclei pellet was resuspended in an appropriate volume of chilled 1× Nuclei Buffer (2000153; 10x Genomics), assuming targeted nuclei recovery of 10,000. If cell debris and large clumps were observed, nuclei suspension was passed through a cell strainer. The concentration and quality of nuclei were assessed using 0.4% Trypan blue staining checked on the automated counting machine Cellometer Mini (Nexcelom, Lawrence, MA, USA), as well as using a standard hemocytometer and light microscopy. Nuclei were immediately used to prepare scATAC-seq libraries.

**Library preparation for 10x Genomics single-cell ATAC sequencing.** The scATAC-Seq libraries were prepared using the 10x Single Cell ATAC Solution (https://www.10xgenomics.com/products/single-cell-atac/), according to the manufacturer's protocol "CG000168 Chromium Single Cell ATAC Reagent Kits Rev B (v1 Chemistry), (10x Genomics, Pleasanton, CA, USA)". The main steps of scATAC-seq library preparation include (1) nuclei transposition, (2) GEM generation and barcoding, (3) post GEMs incubation cleanup and (4) library construction. Briefly, nuclei suspension targeting recovery of 10,000 nuclei per sample were mixed and incubated (37 °C for 60 min) with transposition mix, allowing the transposase to enter the nuclei and fragments the DNA within open-chromatin regions. Also, during this step, adapter sequences are added to the ends of the DNA fragments. Next, the transposed nuclei were mixed with master mix containing barcoding reagents and loaded onto a Chromium Chip E along with Chromium Single Cell ATAC Gel Beads v1 and Partitioning Oil. The nanoliter-scale Gel Beads-in-emulsion (GEMs) were generated using 10x Chromium Controller. The GEMs were captured and incubated (Step 1: 72 °C for 5 min, Step 2: 98 °C for 30 s, Step 3: 98 °C for 10 s, Step 4: 59 °C for 30 s, Step 5: 72 °C for 1 min, Hold: 15 °C; Steps 3–5 were performed in total of 12 cycles) forming 10× barcoded single-stranded DNA. Next, the GEMs were broken, and pooled fractions were recovered, followed by the post GEM-RT Cleanup. Further on, the generated barcoded amplification product was mixed with the sample index PCR mix and incubated (Step 1: 98 °C for 45 s, Step 2: 98 °C for 20 s, Step 3: 67 °C for 30 s, Step 4: 72 °C for 20 s, Step 5: 72 °C for 1 min, Hold: 4 °C; Steps 2–4 were performed in total of 11 cycles) to generate the indexed scATAC libraries. Next, the double-sided size selection using SPRIselect (Beckman Coulter) was performed along with elution of final libraries. Next, the libraries were checked for the fragment size distribution using Agilent 4200 Tape Station HS D1000 Assay (Agilent Technologies) and quantified with Qubit Fluorometric dsDNA Quantification kit (Thermo Fisher Scientific, Waltham, MA, USA). Each of scATAC libraries contain the P5 and P7 sequences used in Illumina® bridge amplification and contain the unique sample indexes. The libraries were sequenced at the ATGC MDACC core facility, with each library on a separate lane of HiSeq4000 flow cell (Illumina), with the sequencing targeting above the minimum of 25,000 read pairs per nucleus sequencing depth, format of 100nt and parameters (Read 1—50 cycles, Read 2—50 cycles; with exception while sequencing together with scRNA libraries Read 1—100 cycles, Read 2—100 cycles).

**scATAC-seq data processing, quality control, and data analysis.** Raw sequencing reads were mapped to the human hg19 reference genome (GRCh37, 10X Genomics), and quantified using cellranger-atac count function with default parameters (Cell Ranger ATAC, version 1.1.0, 10X Genomics). R package Signac was used for scATAC-seq data processing and analysis[26]. Briefly, cells were filtered based on a combination of metrics: (1) the transcription start site (TSS) enrichment score no less than 1, (2) the nucleosome signal (NS) score lower than 4, (3) the total number of fragments in peaks kept between 10th and 90th percentile per sample, (4) ratio of reads mapped to peaks higher than 15, and (5) fraction of reads mapped to blacklisted genomic locations lower than 5%. Cells satisfying all of the above five measures were kept for further analysis.

Peaks were called using MACS2 algorithm[94] and merged to yield a reduced set of regions, which was later used to generate a combined sparse count matrix across all the samples. In search of integration anchors across all the samples, reciprocal latent semantic indexing (LSI) projection[95] was applied to project our data into a shared low-dimensional space[96]. We then integrated the low-dimensional cell embeddings across samples. Dimensionality reduction was performed using Iterative Latent Semantic Indexing (LSI) projection[95] and single-cell embeddings were generated using Uniform Manifold Approximation and Projection (UMAP; nNeighbors = 40; minDist = 0.1). Further downstream analyses, including marker peak identification, gene activity calculation, peak-linked genes identification and reference peak set enrichment with ChromVAR deviations, were performed using Signac default functions. To map peak-related genes, gene with the highest fold change will be applied to represent the peak, if multiple gene promoters were mapped next to the same peak.

To confirm the consistency of cell label transfer, we employed a second scATAC-seq processing pipeline named archR[28]. Briefly, raw fragments files were loaded into ArchR to generate Arrow files. Cells were filtered based on minimal TSS enrichment score of 8 and minimal fragment number of 1000. Doublets were inferred and removed using standard ArchR parameters. Dimensionality reduction was performed using Iterative Latent Semantic Indexing (LSI; $N = 3$), and single-cell embeddings were generated using Uniform Manifold Approximation and Projection (UMAP; nNeighbors = 40; minDist = 0.1). Peaks were called using MACS2 algorithm. Further downstream analysis, including cell label predictions, gene score calculation, marker peak identification, and motif enrichment analysis with ChromVAR were performed using ArchR default analytic functions. Further comparison with Signac pipeline was carried out using the same set of cells called by both pipelines. Percentage of cells assigned with the same cell types from both pipelines were calculated.

**Defining scATAC-seq cluster identity with scRNA-seq data.** Our previously published scRNA-seq profiles of the same bone marrow aspirates were used as reference to map scATAC-seq clusters into defined cell types. Briefly, to help interpret the scATAC-seq data, we aimed to classify cells based on scRNA-seq data from the same biological system using R package Signac[26]. We utilize methods for cross-modality integration (function *FindTransferAnchors*) and label transfer (function *TransferData*) to identify shared correlation patterns in the gene activity matrix calculated from scATAC-seq data and scRNA-seq dataset to identify matched biological states across the two modalities. Binary classes of malignant cells and tumor microenvironment (TME) cells (i.e., normal cells) were first predicted, followed by detailed cell-type predictions in TME cells. To yield a better separation between AML malignant cells and TME cells, different subsets of peaks were utilized to conduct label transfer from reference scRNA-seq data to scATAC-seq data. Open-chromatin peaks were mapped into three relevant categories, i.e., promoter, proximal, and distal enhancer regions. Genomic annotations for the above three regulatory regions were downloaded from the ENCODE Encyclopedia (Version 5)[97]. In combination with SC3 consensus clustering stability[27], the peak set with the highest stability score was applied to generate the final cell labels for both broader and detailed cell types.

**Defining cell states of AML malignant cells in scATAC-seq and scRNA-seq data.** Signature peak sets unique to ten normal hematopoietic cell types were downloaded from ref. [29], wherein both bulk and CD34+ sorted bone marrow cells were profiled. Signac function chromVAR was applied to calculate the deviations in chromatin accessibility across the input set of peak regions. A deviation score cutoff of 1.96 was used to assign cell states to AML malignant cells in scATAC-seq data. AML cells were either assigned to one unique cell state, none cell state, or labeled as a hybrid. To assign cell state labels to AML malignant cells in scRNA-seq data based on their closest hematopoietic counterpart, a KNN classifier was employed to label AML cells based on the ten nearest-neighbors using R package Symphony[39]. For analysis of cell composition within each AML sample, only projected labels

assigned with at least 80% probability (agreement between 8 and 10 neighbors) to at least 5% of cells within any given sample were retained for further analysis.

Commonly shared malignant cell states between scATAC-seq and scRNA-seq were kept for further differential analysis. Peaks were linked to genes based on the distance to their promoters using function *LinkPeaks* in Signac to find peaks that were significantly correlated with the expression of nearby genes. For each peak-gene pair, differentially accessible regions (DARs) were calculated by comparing cells with malignant cell states assigned to their normal counterparts using scATAC-seq data. Differentially expressed genes (DEGs) were then calculated similarly using scRNA-seq data. The significance level for both DARs and DEGs were set as FDR < 0.05. Peak-gene pairs were dropped from further analysis if one component was not called as a differential event (FDR < 0.05). Cell states unique DARs were calculated by comparing cells with cell states assigned to cells without any cell states assigned.

Gene set enrichment analysis was performed using all the available gene sets from the Molecular Signatures Database (MSigDB)[47], by excluding gene set terms starting with "MODULE_" and "GNF2_" due to minimal meaningful information from their systematic names.

**Trajectory analysis**. Trajectory analysis using merged malignant cells was performed using Monocle3[98]. Briefly, malignant cells with stemness state, such as ActHSPC and LTHSPC were combined as the initial cell populations. Pseudotime for each trajectory was thus computed for each cell along the trajectories. Peaks contributing to each cellular trajectory was also extracted for downstream analysis.

To incorporate the longitudinal design, trajectory analysis per sample was performed using ArchR default function. Briefly, confusion matrix was first created to list the cluster proportions across samples from different timepoints. The proportion of each cluster within sample from timepoint A was then used to order the clusters with the highest proportion being the first while least proportion being the last. The order was then used by ArchR default trajectory analysis function to generate the trajectories. Finally, manual inspection was performed to ensure key markers were following the trajectory correctly.

**Survival analysis**. The RPKM TCGA-LAML expression profiles ($n = 170$) were downloaded (https://gdc.cancer.gov/about-data/publications/#/?groups=TCGA-LAML&years=&order=desc)[71], together with the survival data from Bioconductor RTCGA.clinical ("patient.vital_status"). Next, we took all genes that were linked to any of the HSPC-, MEP-, monocyte- and B-cell-like AML cells, and computed row-wise z scores for each gene. AML patients were scored using each set of gene signatures with R package GSVA[99]. Patients were assigned to one of the four cell states when their scores were more than 0. We then split patients into two groups by extracting patients assigned to either HSPC- and MEP-like or monocyte- and B-cell-like AML cells. The significance *P* value was calculated using the R package survival. We plotted the Kaplan–Meier curve using the R package survminer. Our previous treatment-naive AML cohort ($n = 90$) with prognosis information was applied as an independent validation set for survival analysis[6].

**Statistics and reproducibility**. Wilcoxon rank-sum test (two-sided) was applied in scRNA-seq marker gene analysis and scATAC-seq marker peak analysis. Hypergeometric test was used for gene set enrichment analysis. Log-rank test was applied to test the difference between survival groups. False discovery rate (FDR) adjustment was performed using the Benjamini–Hochberg algorithm. Survival analysis was replicated using an independent dataset to show reproducibility.

**Reporting summary**. Further information on research design is available in the Nature Portfolio Reporting Summary linked to this article.

## Data availability
All raw data have been uploaded to European Genome-Phenome Archive (EGA) (Accession # EGAD00001007675; https://ega-archive.org/dacs/EGAC00001002085) and are publicly available. All processed data, processing, and analysis scripts are available upon reasonable request.

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

## Acknowledgements

We would like to acknowledge the Advanced Technology Genomics Core (ATGC) at MD Anderson for sequencing and technical support. The work was supported in part by Leukemia Research Foundation Award, Institutional Start-Up Funds, and Conquer Cancer Foundation Young Investigator Award to H.A.A.; CA016672 and NIH 1 S10OD024977-01 to Advanced Technology Genomics Core (ATGC); the Dick Clark Immunotherapy Fund for N.D.; and APOLLO, Welch Foundation and CPRIT grants to A.F.; H.F. and Z.Z. were supported by the Cancer Prevention and Research Institute of Texas (CPRIT RP180734). Z.Z. was partially supported by NIH grant R01LM012806.

## Author contributions

H.A.A. conceived and designed the study and conducted the analysis. N.D. and A.F. co-designed the study with H.A.A. H.F., F. W., A.Z., B.W., and A.M. performed bioinformatic data analysis and data interpretation. S.L., K.C. L.W., D. H., J.D., Z.Z., and A.K.J. assisted with data interpretation. K.T. and P.B. performed library preparation and experimental optimization: K.R. supervised library preparation. F.Z.J. performed hematopathologic evaluation. H.A.A., H.F., and F.W. wrote the manuscript, with all authors contributing to the writing and providing feedback.

## Competing interests

The authors declare no competing interests.
