## [Peer Review File · Communications Biology]

Reviewers' comments:

Reviewer #1 (Remarks to the Author):

Summary and general comments

In this study, Fen, Wang and colleagues investigate the chromatin accessibility landscape of bone marrow (BM)-derived cells from AML patients primarily refractory or relapsed after first therapy at single-cell resolution. The study includes a follow-up sampling from time of therapy resistance (A) to initiation of an alternative azacitidine+nivolumab therapy (timepoints B and C) to further investigate the dynamics of chromatin reprogramming upon treatment. The study evidences a heterogeneous lineage architecture in the BM of AML patients resistant to first line treatment, with malignant cells enriched for hematopoietic and stem-like phenotypes but also primed for differentiated myeloid, erythroid and lymphoid lineages. Such a heterogeneity is kept after therapy.

While the study itself deserves consideration to be published in the present journal, I am afraid a major revision of the analysis approach and data presentation, as well as interpretation, should be performed by authors. Below my open points and suggestions:

Major revisions

- 1) The study design includes 8 AML patients relapsed/refractory to (7/8) hypomethylating therapies, admitted to a ICB-therapy trial. Though the information on previous therapies is provided in the scRNA-seq BM dataset published on Nature Communications in 2021 [7], it should be specified also here. Moreover, a clear overview on patients' response to ICB-therapy (as provided in [7]) - R, NR, SD classification - should be provided in the Main. Paragraph 1, very beginning lines 94-98, needs to be extended with these information plus a clear recap of sampling time for all patients. Would be also helpful to summarize the overall design in an additional introductory Figure.
- 2) The analysis only superficially refers to the previous scRNA-seq and TCR repertoire datasets. Dedicated comparative analyses and discussion should be included along the manuscript when describing ATAC or integrated findings. The presented integration strategy is superficial and very biased, while it should be thoroughly investigated following gold standard practices in the sc multiomics field to get the complete picture on gene regulation.
- 3) The overall message of the paper is unclear compared to the longitudinal design of the study. Findings mostly relate to the time of therapy resistance (A) with insufficient focus on the disease evolution following ICB-therapy. Moreover, if major findings relate to timepoint A (time of relapse/refractoriness), proper controls (patients which responded to previous therapy - hypomethylating agents? - or, even better, matched control samples for relapsed patients) are missing to make such findings sound enough.

More technical points in detail:

- 4) Lines 94-98: Please detail the cohort as previously suggested and add a study design Schema in Figure 1.
- 5) Lines 103-117: Figure 1 B and C are not particularly informative and could be moved to Supplementary figures, while a clear UMAP annotation of cell types and states should be added instead to the label transfer approach worked successfully. Concerning the two healthy controls, were these processed together with the other samples and do control cells merge with normal cells? In general, did you check and in case correct for any batch effect?
- 6) Line 128: related to Figure 1E, information of DE call is missing in the Methods and should be added. DE call is performed on the previous scRNA-seq dataset and these genes were plotted for chromatin accessibility. How were peaks selected for plotting when multiple peaks mapped a gene? Are these gene activity scores, promoter mapping peaks, other? This should be specified in the main

and also in the Methods. Being the annotation driven by scRNA-seq data, would be more helpful, to assess the goodness of label transfer annotation, to check for deduced marker peaks per each scATAC-seq cluster of interest and plot their accessibility, highlighting key associated genes where present.

7) Line 138: A UMAP of annotated cell states in malignant cells should be added in Figure 2. Figure S2 A plots it across patients, is it correct (I don't see any reference in the text)? How do the big unlabeled clusters behave like in the original UMAP? Did you try to manually annotate major clusters using ATAC-deduced markers, and are you sure you can exclude batch effects or tried correction for those in present?

8) Line 138-145: Figure 2A and 2B are mixed up in the text. Related to Fig 2A, could you better speculate on the number of malignant cells obtained per patient very low in 2 cases?

9) Line 152 on: Could you add some statistics to cell states frequencies (i.e. R, NR, SD patients)? If you consider the 22 samples, this should be possible.

10) Line 152 on: How do these cell states frequencies observations relate to scRNA-seq data? This parallelism should be tackled overall more in the manuscript.

11) Line 165: Figure 2C, could you please comment on your decision to perform DAR analysis on malignant cells cell state-wise assigned vs unassigned? Why this and not malignant assigned cell state vs all the other malignant assigned states? Unassigned cells are not easy to characterize in terms of composition.

12) Line 193: Integration needs to be better defined in the methods. Is this only a comparison between RNA-seq and ATAC-seq regulation? If yes, how did you deal with multiple peaks associated to single genes to check the parallel regulation in transcription and chromatin accessibility?

In general, a more refined and thorough omic integration should be performed, which would help to correlate this data to the previous RNAseq findings, poorly addressed along the manuscript.

13) Line 194-198: I don't understand how you ended up with DAR-DEG pairs. Did you consider DARs and associated genes and called them DEGs and vice versa? It is not clear what is truly significantly different among all these DARs and DEGs, like this, it seems that all DEGs are also DARs which is unlikely. Also, concordant pairs are truly both DE and DA? There is a difference between checking for regulation and checking for statistically significant regulation. Should be better specified if DEGs are only DAR-associated genes and what finally contributed to signatures.

14) Line 198, Figure 3: Would be more transparent and informative to show the heatmap on concordant genes in a non-binarized form (i.e. plotting a log₂FC).

15) Line 234: not sure I understood what is exactly the cell input for the pseudotime analysis? Could you provide also a cell state-annotated reference that we can use as a guide to check for pseudotime evolution?

Minor revisions

16) Please add some graphics annotation or color code to plots at least in main figures (i.e. timepoint, responders).

17) Figure S2 is not mentioned in the second section of Results but only in next paragraph (and looks mixed up with S3).

18) Line 196: rephrase "correlated"

Reviewer #2 (Remarks to the Author):

Fan et al., have profiled single-cell open-chromatin of 22 AML bone marrow aspirates from 8 patients at the time of therapy resistance and subsequent therapy to characterize their lineage landscape. They claim to have identified multiple kinds of therapy-resistant AML cells that were primed for stem and progenitor lineages. They have further tried to identify their states; quiescent, activated, or late stem cell/progenitor. They claim that many resistant AML cells were also composed of cells primed for differentiated myeloid, erythroid, and lymphoid lineages. Hence their message is that that therapy-resistant AML cells also constitute of continuum of differentiated cellular lineages. Further they argue that heterogeneity in lineages likely contributes to their therapy resistance by harbouring different degrees of lineage-specific susceptibilities to therapy. They only explain the severity of the problem rather than suggesting any solution. Such severity of problem for leukemia has also been shown by a few previous studies.

Overall, the data-set of drug resistant version is unique which may help other scientist to develop therapeutic approaches. However authors themselves have not analysed it thoroughly for better inferences which could have been possible. In addition I have a few more concerns :

1) Author should verify the label transfer to single-cell ATAC-seq from scRNA-seq profile using 1-2 other tools also. As there have been two publications which have shown that label transfer by integrative methods could be wrong (Mishra et al., Genome research, 2023 ; Leuken et al., <https://www.nature.com/articles/s41592-021-01336-8>). One should also verify by annotating individual single cells using other approaches also (https://cdn.10xgenomics.com/image/upload/v1660261285/support-documents/CG000234_TechnicalNote_CellTypeAnnotationUsingATAC_RevB.pdf). Current integrative methods for scATAC-seq are not totally reliable. Though authors have shown proof using peaks specific to group of cells. Complete faith on Signac is not recommended.

2) Even for trajectory analysis using monocle3 is doubtful if it relies on gene-activity score as it has been warned by Leuken et al, Nature methods that gene-activity score might not be correctly representing the ATAC-seq profile. One should verify using two independent methods.

3) Identification of normal and malignant cells is doubtful especially for drug-resistant version. Kindly elaborate more in supplementary text, highlighting robustness of the analysis.

4) HOXA family transcription factors are anyway involved in early haematopoietic cells. How the authors can say that <https://www.ncbi.nlm.nih.gov/pmc/articles/PMC6081605/>) they have made a contribution. Authors could check the gene-activity looking at promoters in different groups of cells and make more specific, novel and helpful conclusion.

5) The presentation of enriched pathway in figure 3 can be improved. Kindly move this to supplementary figure and come up with better presentation to highlight more relevant terms, properly discussed in text.

6) Can author's put some text in discussion how the malignant progenitor cells are different from normal stem-like progenitor cells.

Minor comments

In Figure-1 caption, kindly put full abbreviation of TME.

Reviewer #3 (Remarks to the Author):

In this study, the authors investigated the heterogeneous lineage composition of AML cells at time of therapy resistance using single-cell ATAC-seq assay. They found that therapy resistant AML cells were primed for stem and progenitor lineages as well as various differentiated lineages. They further claimed the lineage composition revealed by the chromatin accessibility features could be used for prognosis prediction in TCGA AML cohort.

In general, this study provides a new view of lineage composition of therapy resistant AML cells by profiling the chromatin accessibility in single cells comparing with previous reports by single cell transcriptomes. These data could be a nice resource and of great interest to researchers in AML field. The analysis in this study is quite clear and solid.

There are a few questions the authors need to address:

- 1, What are the major difference in term of chromatin accessibility between each AML malignant lineages and their normal counterpart?
- 2, Are there many repeat element regions being activated in therapy resistant AML cells?

Response to the comments of Reviewer 1

Manuscript ID: COMMSBIO-23-0253

Title: Single Cell Chromatin Accessibility Profiling of Acute Myeloid Leukemia Reveals Heterogeneous Lineage Composition at Time of Therapy-Resistance

We very much appreciate reviewers' careful and thorough reading of our manuscript and their thoughtful comments, which have helped strengthen and clarify our work. Below, for your convenience, we quote the reviewers' original comments in *"italic blue"*, respond to each point in regular font, and include the most relevant revised text here in *"quoted red"*. Manuscript was revised in response to reviewers' comments with tracked changes.

"In this study, Fen, Wang and colleagues investigate the chromatin accessibility landscape of bone marrow (BM)-derived cells from AML patients primarily refractory or relapsed after first therapy at single-cell resolution. The study includes a follow-up sampling from time of therapy resistance (A) to initiation of an alternative azacitidine+nivolumab therapy (timepoints B and C) to further investigate the dynamics of chromatin reprogramming upon treatment. The study evidences a heterogeneous lineage architecture in the BM of AML patients resistant to first line treatment, with malignant cells enriched for hematopoietic and stem-like phenotypes but also primed for differentiated myeloid, erythroid and lymphoid lineages. Such a heterogeneity is kept after therapy.

While the study itself deserves consideration to be published in the present journal, I am afraid a major revision of the analysis approach and data presentation, as well as interpretation, should be performed by authors. Below my open points and suggestions:"

We much appreciate your assessment on our manuscript, and valuable suggestions which have helped enhanced our manuscript significantly.

"Major revisions

1) The study design includes 8 AML patients relapsed/refractory to (7/8) hypomethylating therapies, admitted to a ICB-therapy trial. Though the information on previous therapies is provided in the scRNA-seq BM dataset published on Nature Communications in 2021 [7], it should be specified also here. Moreover, a clear overview on patients' response to ICB-therapy (as provided in [7]) - R, NR, SD classification – should be provided in the Main. Paragraph 1, very beginning lines 94-98, needs to be extended with these information plus a clear recap of sampling time for all patients. Would be also helpful to summarize the overall design in an additional introductory Figure."

Thank you for your helpful suggestion. We now added one main introductory figure panel for a detailed recap of data applied in our current study and a summary of overall design.

Added Figure 1A.

Figure 1A. An introduction of the recruited cohort for our study which included 8 patients. Longitudinal bone marrow samples were extracted per patient for sequencing. First batch of samples were profiled using paired scRNA-seq and TCR-seq, with a highlight of CD8 T cells (published), while second batch of samples were profiled using scATAC-seq and centered on the analyses of malignant cells by including gene expression as a complement.

Added Results:

“We performed scATAC-seq profiling on 22 whole bone marrow aspirates collected at different treatment timepoints from 8 AML patients who received prior therapies and relapsed or were primary refractory to PD-1 blockade, hereafter referred to as therapy-resistant AML. All patients were treated on protocol [25] at time of therapy-resistance (pretreatment = timepoint A) with azacitidine and nivolumab (following treatment = timepoints B and C). Briefly, our cohort had a median age of 73 years (range 64-88) prior to receiving PD-1 blockade therapy. While on the treatment, 3 patients (PT1-3) were responders (R); 3 patients (PT4-6) were non-responders (NR) and 2 patients (PT7-8) showed stable disease (SD) (Figure 1A). Combining our published scRNA-seq gene expression data using the same cohort [7], we aim to reveal the chromatin landscape in these AML patients with a focus on malignant cells.”

“2) The analysis only superficially refers to the previous scRNA-seq and TCR repertoire datasets. Dedicated comparative analyses and discussion should be included along the manuscript when describing ATAC or integrated findings. The presented integration strategy is superficial and very biased, while it should be thoroughly investigated following gold standard practices in the sc multiomics field to get the complete picture on gene regulation.”

We thank the reviewer for this critical comment. In response to your critique, we added more details in Discussion regarding chromatin accessibility and their impact on gene regulation. The main topic of our current study is to dissect the landscape of chromatin accessibility in AML patients at the time of therapy resistance. Gene expression profile is served as complementary information (already published) for an easier explanation of

functions enriched by peak-associated genes since genomic regions are difficult to interpret. As far as integration between chromatin accessibility and gene expression to reveal gene regulation, there is no such gold standard practice in the field as far as we know. On top of that, open chromatin regions represent the potential of genomic regions open for regulations, which could be either positive or negative. And that is why the genome-wide correlation between chromatin accessibility and gene expression is relatively low (Nair *et al.*, 2021). For example, DNA methylation could emerge within open chromatin regions, which has been shown to silence gene expression when occur within their promoter regions. Using one layer of epigenetic mark, i.e., chromatin accessibility, to reveal the complete picture for gene regulation is unrealistic and inaccurate, especially difficult when scRNA-seq and scATAC-seq data are not profiled using the exact same single cells. Gene regulation is a complex process which involves the interplay of multiple layers of genomic and epigenomic regulations (Ricketts *et al.*, 2018). ATAC-seq enables the genome-wide identification of transcription factor binding events that orchestrate gene expression (Corces *et al.*, 2018). Therefore, we integrated gene expression with our identified peak signatures when necessary. We agree with your view, that this is one of our limitations. For future experimental design, we will keep in mind to apply chromatin accessibility and gene expression profiling on the same single cells to yield the most optimal integration results.

Added Discussion:

“Specifically, we have observed a discrepancy in regulatory potential between open chromatin-marked signature regions and their linked genes. Similar phenomena regarding the low correlations between open chromatin signals and their linked gene expression profiles have been studied using rat adipose and muscle tissues, with the highest correlation nearing 0.4 within promoter regions in muscle [74]. **ATAC-seq enables the genome-wide identification of transcription factor binding events to orchestrate gene expression [81]. Since gene regulation is a complex process which involves the interplay of multiple layers of genomic and epigenomic regulations [82], it is especially difficult to reveal the complete picture for gene regulation when scRNA-seq and scATAC-seq data are not profiled using the exact same single cells, with one layer of epigenetic mark, i.e., chromatin accessibility.**”

“3) The overall message of the paper is unclear compared to the longitudinal design of the study. Findings mostly relate to the time of therapy resistance (A) with insufficient focus on the disease evolution following ICB-therapy. Moreover, if major findings relate to timepoint A (time of relapse/refractoriness), proper controls (patients which responded to previous therapy – hypomethylating agents? – or, even better, matched control samples for relapsed patients) are missing to make such findings sound enough.”

Thank you for your comment, and we agree with you that the longitudinal perspective is not well-addressed patient-wise. Despite we agree with you that adding proper control samples will make our current findings more sound, this small clinical trial has been closed currently. Adding matched control samples as you suggested won't be feasible at this

point. Therefore, we decided to add in patient-wise trajectories to show the potential evolutionary path from timepoint A to C if data permits.

Added Figure S7.

Figure S7. Pseudotemporal analysis reveals chromatin accessibility of canonical transcription factor motifs. (A) UMAP embeddings showing the trajectories of all 8 patients. (B) Line plots showing the pseudotemporal accessibility changes (based on gene scores) of genes associated with stemness (CD34) as well as those of lineage commitment related genes such as CD14. (C) Line plots showing the pseudotemporal signature Z score changes of various accessibility signatures which was based on sorted well-defined hematopoietic cells from different hematopoietic lineage states. (D) Heatmaps showing the pseudotemporal motif enrichment of key transcriptional factors for PT3 (Responder), PT4 (Non-responder) and PT7 (Stable Disease).

Added Results section.

“We next leveraged the longitudinal inferred trajectory with supervised pseudotime ordering to understand AML longitudinal topology and identified variability in lineage trajectories following treatment (Figure S7A). Trajectory analysis revealed dynamic changes of accessibility of genes associated with stemness (CD34, MLLT3, PROM1, GATA2, FLI1) [70] as well as those of lineage commitment related genes such as CD14, MPO, ALAS2, MPEG1 and CD19 (Figure S7B). We also performed hematopoietic accessibility signatures analysis and overlaid the enrichment score along the trajectory (Figure S7C). We focused our analysis on patient 3 who acquired a cytogenetic deletion in chromosome 7 at timepoint C; patient 4 who had an evolution of CD34- CD14+ monocytic cells at time of disease resistance (timepoint B) and patient 7 who had stable disease for >1 year while on therapy (timepoints B and C). Patient 3 and patient 7 showed increasing primitive stem cell signatures/genes along the trajectory while patient 4 showed increasing monocyte signature/genes and decreasing primitive stem cell signatures/genes (Figure S7B-S7C). Motif enrichment along trajectory also showed consistent results. Specifically, Patient 3 showed transitions from differentiation-related motifs (such as CEBP family) to progenitor related motifs (such as GATA family) (Figure S7D), while patient 4 showed an opposite pattern (transitioned from GATA enriched to CEBP and SPI1 enriched) (Figure S7E). Patient 7 also showed motif enrichment of LMO2, which is an essential transcriptional factor for primitive hematopoiesis, at the late stage along the trajectory (Figure S7F). These findings suggest dynamic changes in epigenetic accessibility following treatment that could modulate cellular states.”

Added Methods section.

“To incorporate the longitudinal design, trajectory analysis per sample was performed using ArchR default function. Briefly, confusion matrix was first created to list the cluster proportions across samples from different time points. The proportion of each cluster within sample from time point A was then used to order the clusters with highest proportion being the first while least proportion being the last. The order was then used by ArchR default trajectory analysis function to generate the trajectories. Finally manual inspection was performed to ensure key markers were following the trajectory correctly.”

“More technical points in detail:

4) Lines 94-98: Please detail the cohort as previously suggested and add a study design Schema in Figure 1.”

Thank you for your informative suggestion. We have added the cohort details (Figure 1A) and study design as you suggested in Figure 1 now.

“5) Lines 103-117: Figure 1 B and C are not particularly informative and could be moved to Supplementary figures, while a clear UMAP annotation of cell types and states should be added instead to the label transfer approach worked successfully. Concerning the two healthy controls, were these processed together with the other samples and do control

cells merge with normal cells? In general, did you check and in case correct for any batch effect?"

As you suggested, we moved Figure 1B and 1C to supplementary Figure S1 (now Figure S1A and S1E). And yes, the batch effect is well handled through sample integration process using common anchors in R package Signac (Stuart *et al.*, 2022). No batch effects were observed based on dimension reduction analysis, either. During scATAC-seq profiling process, all samples are handled together under the 10x Genomics guidance. Cells from healthy controls are not merged with normal cells within tumor microenvironment. The main theme of our current study is to focus on malignant AML cells; therefore, the two normal samples were excluded in our current Figure 1A UMAP. For a global trend comparison, these two normal samples are included in Figure 1B to show healthy baseline distribution (now Figure S1A). We added a UMAP with detailed cell type annotated to show the result of label transfer using our published reference scRNA-seq data (Figure 1C).

Added Figure 1C.

Added Results.

“By applying Signac pipeline [26], a total of 59,321 mononuclear bone marrow individual cells passed quality control after excluding 2 low-quality samples, and were grouped into 2 broader clusters, i.e., AML malignant cells and normal cells in the tumor microenvironment (TME) cells (Figure 1B), with granular cell labels shown in Figure 1C.”

“6) Line 128: related to Figure 1E, information of DE call is missing in the Methods and should be added. DE call is performed on the previous scRNA-seq dataset and these genes were plotted for chromatin accessibility. How were peaks selected for plotting when multiple peaks mapped a gene? Are these gene activity scores, promoter mapping peaks, other? This should be specified in the main and also in the Methods. Being the annotation

driven by scRNA-seq data, would be more helpful, to assess the goodness of label transfer annotation, to check for deduced marker peaks per each scATAC-seq cluster of interest and plot their accessibility, highlighting key associated genes where present.”

Thank you for your comments. Figure 1E is a heatmap of all the peaks identified, not just DE or promoter mapped peaks, with row clustered across all the major cell types. Most relevant marker peak-specific genes were labeled in the column. In the heatmap, each row is a major cell type, while each column is a peak. Accessibility values, not gene activity scores, were z-scored and averaged within each cell type for the heatmap visualization. Figure legend are revised accordingly to reflect more detailed information.

If multiple gene promoters were mapped next to the same peak, gene with the highest fold change will be applied to represent the peak. Details regarding this information is now added in the Methods.

Added Methods.

“Further downstream analyses, including marker peak identification, gene activity calculation, peak-linked genes identification and reference peak set enrichment with ChromVAR deviations, were performed using Signac default functions. **To map peak-related genes, gene with the highest fold change will be applied to represent the peak, if multiple gene promoters were mapped next to the same peak.**”

Cell type-specific peak signatures were identified first, and then their linked most relevant genes were labeled for an easy interpretation. We verified our label transfer with a different pipeline named archR, where cluster-based markers (new Figure S2C-S2E) are shown based on gene activity score to accompany Figure 1.

Added Figure S2

Figure S2. Verification of cell label transfer. (A) Heatmap showing the consistency of major cell type calling between two different scATAC-seq processing pipelines. Percentage of overlapped cell type are labeled on the heatmap. (B) UMAP of scATAC-seq cell types predicted using archR pipeline. (C) Heatmap showing the marker genes (based on gene activity scores) across different cell types. (D) UMAP embeddings of all cells showing mapped gene expression by integrating scRNA-seq data (upper panel), and gene activity scores predicted using scATAC-seq data (lower panel) of CD3E, CD8B and HBB. (E) Heatmap showing the motif enrichment across different cell types.

Added Methods.

“To confirm the consistency of cell label transfer, we employed a second scATAC-seq processing pipeline named archR [28]. Briefly, raw fragments files were loaded into ArchR to generate Arrow files. Cells were filtered based on minimal TSS enrichment score of 8 and minimal fragment number of 1,000. Doublets were inferred and removed using standard ArchR parameters. Dimensionality reduction was performed using Iterative Latent Semantic Indexing (LSI; N=3) and single cell embeddings were generated using Uniform Manifold Approximation and Projection (UMAP; nNeighbors = 40; minDist = 0.1). Peaks were called using MACS2 algorithm. Further downstream analysis, including cell label predictions, gene score calculation, marker peak identification, and motif enrichment analysis with ChromVAR were performed using ArchR default analytic functions. Further comparison with Signac pipeline was carried out using the same set of cells called by both pipelines. Percentage of cells assigned with the same cell types from both pipelines were calculated.”

Added Results.

“By re-applying ArchR pipeline, we were able to verify that more than 90% of the malignant cells called by Signac were consistent with archR predictions (Figure S2A). Cell label annotations of the scATAC-seq clusters by archR uncovered 9 bone marrow cell types of erythroid, lymphoid, and myeloid lineages (Figure S2B). Marker peak analysis revealed distinct peak accessibility across the healthy and leukemic cellular subtypes (Figure S2C), consistent with cell type specific chromatin accessibility profiles

[15, 28, 29, 33]. To further validate the cluster annotation, we visually inspected the tracks of canonical gene markers (Figure S2D) and calculated the lineage-defining gene scores, which represent the overall chromatin accessibility at the gene body and promoter regions [29, 34-36]. For instance, T cells had highest mapped gene expression by integrating scRNA-seq data for CD3E and CD8B, whereas erythroid cells had high gene score for HBB inferred with their epigenetic accessibility (Figure S2D, upper panel), that were consistent with their predicted gene activity scores using scATAC-seq data in the putative cell types (Figure S2D, lower panel). Next, we analyzed the differential accessible chromatin regions for enriched DNA-binding motifs of lineage-specific transcription factors, which correlate with cell identity [37]. Our accessibility profile of transcription factor enrichment scores confirmed canonical transcription factor regulators of the identified cell types (Figure S2E).”

“7) Line 138: A UMAP of annotated cell states in malignant cells should be added in Figure 2. Figure S2 A plots it across patients, is it correct (I don't see any reference in the text)? How do the big unlabeled clusters behave like in the original UMAP? Did you try to manually annotate major clusters using ATAC-deduced markers, and are you sure you can exclude batch effects or tried correction for those in present?”

We apologize for the missing citation of Figure S2A (now Figure 5A). It is now properly cited in the revised manuscript. The malignant cell cluster is surrounded by different cell types from TME cells in the original UMAP. Cell states defined for malignant cells do not warrant clear separations using unsupervised UMAP analysis (added Figure S3), which tend to be continuous (Neftel *et al.*, 2019). Therefore, a star-shaped plot is employed to show how malignant cells are distributed toward different quadrants (i.e., cell states) in a 2D space (Figure 5A).

Added Figure S3.

Figure S3. UMAPs of all the malignant cells highlighted by cells with cell states assigned. UMAP embeddings with monocyte-like (A), MEP-like (B), B cell-like (C), and HSPC-like (D) cell states highlighted.

“8) Line 138-145: Figure 2A and 2B are mixed up in the text. Related to Fig 2A, could you better speculate on the number of malignant cells obtained per patient very low in 2 cases?”

We apologize for the insufficient details in Results.

Added Results:

“The number of malignant cells varied greatly across different samples, ranging from dozens to a couple thousands. This could be partially due to the uniqueness of patients with different responsive states across different sampling timepoints.”

“9) Line 152 on: Could you add some statistics to cell states frequencies (i.e. R, NR, SD patients)? If you consider the 22 samples, this should be possible.”

Thank you for your helpful suggestion. We now added significance tests between R and NR, and between R and SD (Figure S4).

Added Figure S4.

Figure S4. Boxplots showing frequencies of each malignant cell state in patient groups of R (Responders), NR (None responders), and SD (Stable disease). (A) HSPC-like cell state. (B) MEP-like cell state. (C) Monocyte-like cell state. (D) B cell-like cell state.

Added Results.

“We found different degrees of stemness including gradients of long-term (LT) (i.e., quiescent) and activated (ACT) hematopoietic/progenitor stem cell (HSC/HSPC) priming within each patient, and across different patients (Figure S4A). Of note, a considerable proportion of AML cells had chromatin accessibility primed for erythroid (erythroid

progenitors) and myeloid-erythroid progenitors (Figure S4B). Monocyte-like AML malignant cells were more present in responsive patients (PT2 and PT3) when comparing to non-responsive patients and patients with stable disease (from PT5 to PT8) (Figure S4C). On the contrary, megakaryocyte erythroid progenitors (MEP)-like and stem cell-like (LTHSPC and ActHSPC) AML cells were significantly dominated in patients with non-responsive and stable disease (Figure S4A-S4B). Compositions of B cell-like AML cells were not significantly altered across different patient groups (Figure S4D).”

“10) Line 152 on: How do these cell states frequencies observations relate to scRNA-seq data? This parallelism should be tackled overall more in the manuscript.”

Malignant cell states using scATAC-seq were predicted using literature-based signature peak set from 10 major blood cell types. Cell states using scRNA-seq were predicted using their normal counterparts. The cell type annotations in these two references are not exactly the same, which will certainly bias the observation for frequencies, as well as percentage. As we mentioned above, scATAC-seq and scRNA-seq data were not profiled using the same cells, either. We decided to leave out the parallelism but use scRNA-seq data for an easy explanation of peaks.

“11) Line 165: Figure 2C, could you please comment on your decision to perform DAR analysis on malignant cells cell state-wise assigned vs unassigned? Why this and not malignant assigned cell state vs all the other malignant assigned states? Unassigned cells are not easy to characterize in terms of composition.”

Thank you for comments. We are trying to identify differential accessible regions that are unique to each cell states, therefore, the comparisons were made between malignant cells with cell states assigned and those malignant cells without any particular cell state assigned. Unassigned cells are malignant cells in nature.

“12) Line 193: Integration needs to be better defined in the methods. Is this only a comparison between RNA-seq and ATAC-seq regulation? If yes, how did you deal with multiple peaks associated to single genes to check the parallel regulation in transcription and chromatin accessibility?

In general, a more refined and thorough omic integration should be performed, which would help to correlate this data to the previous RNAseq findings, poorly addressed along the manuscript.”

We apologize for the insufficient details in the Methods. It is fully addressed in our revision. Peaks were linked to genes based on the distance to their promoters using function *LinkPeaks* to find peaks that are correlated with the expression of nearby genes. Briefly, for each gene, this function computes the correlation coefficient between the gene expression and accessibility of each peak within a given distance from the gene TSS and computes an expected correlation coefficient for each peak given the GC content, accessibility, and length of the peak. The expected coefficient values for the peak are then used to compute a z-score and p-value.

For significantly correlated peak-gene pairs, we overlapped peaks with cell states-specific DARs using scATAC-seq data and genes with cell states-specific DEGs using scRNA-seq data, separately. Yes, there could be one peak being linked to multiple genes. However, if they were all called as DEGs in the same comparison using scRNA-seq, all of them would be kept for downstream functional enrichment analysis. We did not aim to explain peak-gene pair individual to reveal how each gene was regulated. Instead, we were more interested their functional relevance as gene sets.

Epigenetic landscape profiled using scATAC-seq is our main theme for the paper. Gene expression integration is complementary for a better understanding of the functional relevance of peak signatures. It is not designed as parallel to begin with.

Added Restuls.

“To further characterize the diverse AML cell states, we integrated scATAC-seq and scRNA-seq data to link cell states-associated DARs with DEGs **by comparing malignant cell states with their normal counterparts** (Figure S6, Table S1).”

Added Methods.

“Commonly shared malignant cell states between scATAC-seq and scRNA-seq were kept for further differential analysis. Peaks were linked to genes based on the distance to their promoters using function *LinkPeaks* in Signac to find peaks that were significantly correlated with the expression of nearby genes. For each peak-gene pair, differentially accessible regions (DARs) were calculated by comparing cells with malignant cell states assigned to their normal counterparts using scATAC-seq data. Differentially expressed genes (DEGs) were then calculated similarly using scRNA-seq data. Significance level for both DARs and DEGs were set as $FDR < 0.05$. Peak-gene pairs were dropped from further analysis if one component was not called as a differential event ($FDR < 0.05$).”

“13) Line 194-198: I don’t understand how you ended up with DAR-DEG pairs. Did you consider DARs and associated genes and called them DEGs and vice versa? It is not clear what is truly significantly different among all these DARs and DEGs, like this, it seems that all DEGs are also DARs which is unlikely. Also, concordant pairs are truly both DE and DA? There is a difference between checking for regulation and checking for statistically significant regulation. Should be better specified if DEGs are only DAR-associated genes and what finally contributed to signatures.”

We apologize for the confusions. The definitions of DEGs and DARs are now detailed in Methods.

Added Methods.

“Commonly shared malignant cell states between scATAC-seq and scRNA-seq were kept for further differential analysis. Peaks were linked to genes based on the distance to their promoters using function *LinkPeaks* in Signac to find peaks that were significantly correlated with the expression of nearby genes. For each peak-gene pair, differentially accessible regions (DARs) were calculated by comparing cells with malignant cell states assigned to their normal counterparts using scATAC-seq data. Differentially expressed genes (DEGs) were then calculated similarly using scRNA-seq data. Significance level

for both DARs and DEGs were set as $FDR < 0.05$. Peak-gene pairs were dropped from further analysis if one component was not called as a differential event ($FDR < 0.05$)."

"14) Line 198, Figure 3: Would be more transparent and informative to show the heatmap on concordant genes in a non-binarized form (i.e. plotting a \log_2FC)."

Yes, the original Figure S3 (now Figure S6) is showing heatmaps with fold changes for both gene expression and chromatin accessibility.

"15) Line 234: not sure I understood what is exactly the cell input for the pseudotime analysis? Could you provide also a cell state-annotated reference that we can use as a guide to check for pseudotime evolution?"

The cell input is all the malignant cells for pseudotime analysis. Yes, a cell state-highlighted reference was included as Figure S3 now.

"Minor revisions

16) Please add some graphics annotation or color code to plots at least in main figures (i.e. timepoint, responders)."

Thanks for your comments. Graphics annotation for timepoints and response status are now added as necessary.

"17) Figure S2 is not mentioned in the second section of Results but only in next paragraph (and looks mixed up with S3)."

Sorry for the missing citation. It is now properly cited.

"18) Line 196: rephrase "correlated""

It is now rephrased as "corresponded".

Reference:

- Corces, M. R., Granja, J. M., Shams, S., Louie, B. H., Seoane, J. A., Zhou, W., Silva, T. C., Groeneveld, C., Wong, C. K., Cho, S. W., Satpathy, A. T., Mumbach, M. R., Hoadley, K. A., Robertson, A. G., Sheffield, N. C., Felau, I., Castro, M. A. A., Berman, B. P., Staudt, L. M., Zenklusen, J. C., Laird, P. W., Curtis, C., Cancer Genome Atlas Analysis, N., Greenleaf, W. J. and Chang, H. Y. (2018) 'The chromatin accessibility landscape of primary human cancers', *Science*, 362(6413).
- Nair, V. D., Vasoya, M., Nair, V., Smith, G. R., Pincas, H., Ge, Y., Douglas, C. M., Esser, K. A. and Sealfon, S. C. (2021) 'Differential analysis of chromatin accessibility and gene expression profiles identifies cis-regulatory elements in rat adipose and muscle', *Genomics*, 113(6), pp. 3827-3841.
- Neftel, C., Laffy, J., Filbin, M. G., Hara, T., Shore, M. E., Rahme, G. J., Richman, A. R., Silverbush, D., Shaw, M. L., Hebert, C. M., Dewitt, J., Gritsch, S., Perez, E. M., Gonzalez Castro, L. N., Lan, X., Druck, N., Rodman, C., Dionne, D., Kaplan, A., Bertalan, M. S.,

- Small, J., Pelton, K., Becker, S., Bonal, D., Nguyen, Q. D., Servis, R. L., Fung, J. M., Mylvaganam, R., Mayr, L., Gojo, J., Haberler, C., Geyeregger, R., Czech, T., Slavic, I., Nahed, B. V., Curry, W. T., Carter, B. S., Wakimoto, H., Brastianos, P. K., Batchelor, T. T., Stemmer-Rachamimov, A., Martinez-Lage, M., Frosch, M. P., Stamenkovic, I., Riggi, N., Rheinbay, E., Monje, M., Rozenblatt-Rosen, O., Cahill, D. P., Patel, A. P., Hunter, T., Verma, I. M., Ligon, K. L., Louis, D. N., Regev, A., Bernstein, B. E., Tirosh, I. and Suva, M. L. (2019) 'An Integrative Model of Cellular States, Plasticity, and Genetics for Glioblastoma', *Cell*, 178(4), pp. 835-849 e21.
- Ricketts, C. J., De Cubas, A. A., Fan, H., Smith, C. C., Lang, M., Reznik, E., Bowlby, R., Gibb, E. A., Akbani, R., Beroukhi, R., Bottaro, D. P., Choueiri, T. K., Gibbs, R. A., Godwin, A. K., Haake, S., Hakimi, A. A., Henske, E. P., Hsieh, J. J., Ho, T. H., Kanchi, R. S., Krishnan, B., Kwiatkowski, D. J., Lui, W., Merino, M. J., Mills, G. B., Myers, J., Nickerson, M. L., Reuter, V. E., Schmidt, L. S., Shelley, C. S., Shen, H., Shuch, B., Signoretti, S., Srinivasan, R., Tamboli, P., Thomas, G., Vincent, B. G., Vocke, C. D., Wheeler, D. A., Yang, L., Kim, W. Y., Robertson, A. G., Cancer Genome Atlas Research, N., Spellman, P. T., Rathmell, W. K. and Linehan, W. M. (2018) 'The Cancer Genome Atlas Comprehensive Molecular Characterization of Renal Cell Carcinoma', *Cell Rep*, 23(1), pp. 313-326 e5.
- Stuart, T., Srivastava, A., Madad, S., Lareau, C. A. and Satija, R. (2022) 'Author Correction: Single-cell chromatin state analysis with Signac', *Nat Methods*, 19(2), pp. 257.

Response to the comments of Reviewer 2

Manuscript ID: COMMSBIO-23-0253

Title: Single Cell Chromatin Accessibility Profiling of Acute Myeloid Leukemia Reveals Heterogeneous Lineage Composition at Time of Therapy-Resistance

We very much appreciate reviewers' careful and thorough reading of our manuscript and their thoughtful comments, which have helped strengthen and clarify our work. Below, for your convenience, we quote the reviewers' original comments in *"italic blue"*, respond to each point in regular font, and include the most relevant revised text here in *"quoted red"*. Manuscript was revised in response to reviewers' comments with tracked changes.

"Fan et al., have profiled single-cell open-chromatin of 22 AML bone marrow aspirates from 8 patients at the time of therapy resistance and subsequent therapy to characterize their lineage landscape. They claim to have identified multiple kinds of therapy-resistant AML cells that were primed for stem and progenitor lineages. They have further tried to identify their states; quiescent, activated, or late stem cell/progenitor. They claim that many resistant AML cells were also composed of cells primed for differentiated myeloid, erythroid, and lymphoid lineages. Hence their message is that that therapy-resistant AML cells also constitute of continuum of differentiated cellular lineages. Further they argue that heterogeneity in lineages likely contributes to their therapy resistance by harbouring different degrees of lineage-specific susceptibilities to therapy. They only explain the severity of the problem rather than suggesting any solution. Such severity of problem for leukamia has also been shown by a few previous studies.

Overall, the data-set of drug resistant version is unique which may help other scientist to develop therapeutic approaches. However authors themselves have not analysed it thoroughly for better inferences which could have been possible. In addition I have a few more concerns ."

We much appreciate your assessment on our manuscript, and valuable suggestions which have helped enhanced our manuscript significantly.

"1) Author should verify the label transfer to single-cell ATAC-seq from scRNA-seq profile using 1-2 other tools also. As there have been two publications which have shown that label transfer by integrative methods could be wrong (Mishra et al., Genome research, 2023 ; Leuken et al., <https://www.nature.com/articles/s41592-021-01336-8>). One should also verify by annotating individual single cells using other approaches also (https://cdn.10xgenomics.com/image/upload/v1660261285/support-documents/CG000234_TechnicalNote_CellTypeAnnotationUsingATAC_RevB.pdf).

Current integrative methods for scATAC-seq are not totally reliable. Though authors have shown proof using peaks specific to group of cells. Complete faith on Signac is not recommended."

Thank you for your comments and literature recommendations. Label-transfer in Signac (Stuart et al., 2022) is a great technique to assign cell cluster labels when reference dataset is reliable, particularly useful among the same types of omics data, for instance scRNA-seq data. In our case, despite the scRNA-seq and scATAC-seq data are not

analysis with ChromVAR were performed using ArchR default analytic functions. Further comparison with Signac pipeline was carried out using the same set of cells called by both pipelines. Percentage of cells assigned with the same cell types from both pipelines were calculated.”

Added Results.

“By re-applying ArchR pipeline, we were able to verify that more than 90% of the malignant cells called by Signac were consistent with archR predictions (Figure S2A). Cell label annotations of the scATAC-seq clusters by archR uncovered 9 bone marrow cell types of erythroid, lymphoid, and myeloid lineages (Figure S2B). Marker peak analysis revealed distinct peak accessibility across the healthy and leukemic cellular subtypes (Figure S2C), consistent with cell type specific chromatin accessibility profiles [15, 28, 29, 33]. To further validate the cluster annotation, we visually inspected the tracks of canonical gene markers (Figure S2D) and calculated the lineage-defining gene scores, which represent the overall chromatin accessibility at the gene body and promoter regions [29, 34-36]. For instance, T cells had highest mapped gene expression by integrating scRNA-seq data for CD3E and CD8B, whereas erythroid cells had high gene score for HBB inferred with their epigenetic accessibility (Figure S2D, upper panel), that were consistent with their predicted gene activity scores using scATAC-seq data in the putative cell types (Figure S2D, lower panel). Next, we analyzed the differential accessible chromatin regions for enriched DNA-binding motifs of lineage-specific transcription factors, which correlate with cell identity [37]. Our accessibility profile of transcription factor enrichment scores confirmed canonical transcription factor regulators of the identified cell types (Figure S2E).”

“2) Even for trajectory analysis using monocle3 is doubtful if it relies on gene-activity score as it has been warned by Leuken et al, Nature methods that gene-activity score might not be correctly representing the ATAC-seq profile. One should verify using two independent methods.”

We thank the reviewer for this comment. Our trajectory analysis using monocle 3 does not rely on gene-activity score, it is based on chromatin accessibility. Additionally, we listed below a most recent publication from Granja JM, et al. (Nat Genet. 2021) on benchmarking trajectory analysis (Granja *et al.*, 2021) using a large hematopoiesis dataset. In Extended Data Fig. 10d, the top 3 most applied tools for trajectory analysis yielded nearly identical trajectories using scATAC-seq data. We don't think it is necessary to provide excessive proof of its performance since the monocle 3 has been well benchmarked upon publication and against tools developed later.

Figure link here: <https://www.nature.com/articles/s41588-021-00790-6/figures/14>

“3) Identification of normal and malignant cells is doubtful especially for drug-resistant version. Kindly elaborate more in supplementary text, highlighting robustness of the analysis.”

We apologize for the insufficient details in Methods. Identification of malignant cells and normal cells within the tumor microenvironment (TME) in AML patients is still a challenge, due to the potential continuum of normal cells in TME with malignant cells. We now added more necessary details in the Methods and explained more about its robustness in Results, in combination with a re-analysis using a separate pipeline named archR (please refer to the new Figure S2).

Added Methods.

“Our previously published scRNA-seq profiles of the same bone marrow aspirates were used as reference to map scATAC-seq clusters into defined cell types. Briefly, to help interpret the scATAC-seq data, we aimed to classify cells based on scRNA-seq data from the same biological system using R package Signac [26]. We utilize methods for cross-modality integration (function *FindTransferAnchors*) and label transfer (function *TransferData*) to identify shared correlation patterns in the gene activity matrix calculated from scATAC-seq data and scRNA-seq dataset to identify matched biological states across the two modalities.”

“4) HOXA family transcription factors are anyway involved in early haematopoietic cells. How the authors can say that <https://www.ncbi.nlm.nih.gov/pmc/articles/PMC6081605/> they have made a contribution. Authors could check the gene-activity looking at promoters in different groups of cells and make more specific, novel and helpful conclusion.”

Thank you for your informative suggestion. We apologize if any confusion was raised due to our writing style. Annotations of HOXA family members in the manuscript was served as a proof that our malignant cell state definitions, and their specific chromatin markers and their linked genes are biologically meaningful as verified by known HOXA functions in stem cells and cancer that corresponded to the HSPC-like AML cells in our study. Novel findings are related to the other two poorly characterized cell states, i.e., MEP-like and monocyte-like AML cells. Observations made for these two cell states are unique in our study and not referenced either.

Revised Results section as follows.

“Collectively, HSPC-like AML cell state was **confirmed** and characterized by the significant overlap with previously curated AML signature gene sets (VALK AML CLUSTER 1 and VALK AML CLUSTER 15) [47], the activation of PI3K-Akt signaling pathway (REACTOME PI3K AKT SIGNALING IN CANCER) [48], the enhanced metastasis and mobility by MET signaling pathway (REACTOME SIGNALING BY MET) [49], the dysregulated cell-cell adhesion (GOBP REGULATION OF CELL CELL ADHESION and GOBP HOMOTYPIC CELL CELL ADHESION), the increased therapy resistance (CREIGHTON ENDOCRINE THERAPY RESISTANCE 4) [50] and a putative poor survival marked by the enrichment of TF HOXA4 target genes (HOXA4 Q2) [51].... Interestingly enough, we observed MEP signatures from different fetal organs, such as heart, lung and liver, were significantly enriched in our MEP-like AML gene set (Figure 3D, Golden and Blue branches).”

“5) The presentation of enriched pathway in figure 3 can be improved. Kindly move this to supplementary figure and come up with better presentation to highlight more relevant terms, properly discussed in text.”

Thank you for the comment. we tried multiple ways to list cell states' specific functional terms, however, this is the best presentation we can come up with. This treemap not only lists functional terms, but also organizes them into functional clusters for easy interpretation. Please don't hesitate to let us know if you have a better idea. Functional enrichment using MSigDB gave us a more comprehensive functional map which also contains canonical Gene Ontology and KEGG function terms. We decided to keep this figure as main since it competes our story for the cell states' specific chromatin signatures. We cleaned up the Results section related to this figure by removing low dash (-) in MSigDB functional terms.

“6) Can author's put some text in discussion how the malignant progenitor cells are different from normal stem-like progenitor cells.”

Thanks for your comment. We added more details in the Results section.

“In particular, open DARs **corresponded** with up-regulated DEGs in the same or similar AML cell states across both modalities. **Compared to monocyte- and MEP-like AML cell states, HSPC-like AML cells showed fewer differential events in chromatin accessibility.**”

We also observed that chromatin accessibility tended to be more open when comparing malignant lineages with their normal counterparts (Figure S6). Differential events were then binarized to retain information for differential directions (Figure 3A).”

“Minor comments

In Figure-1 caption, kindly put full abbreviation of TME.”

We spell out TME in Figure 1 caption. TME is the abbreviation of tumor microenvironment.

Reference:

Granja, J. M., Corces, M. R., Pierce, S. E., Bagdatli, S. T., Choudhry, H., Chang, H. Y. and Greenleaf, W. J. (2021) 'ArchR is a scalable software package for integrative single-cell chromatin accessibility analysis', *Nat Genet*, 53(3), pp. 403-411.

Stuart, T., Srivastava, A., Madad, S., Lareau, C. A. and Satija, R. (2022) 'Author Correction: Single-cell chromatin state analysis with Signac', *Nat Methods*, 19(2), pp. 257.

Response to the comments of Reviewer 3

Manuscript ID: COMMSBIO-23-0253

Title: Single Cell Chromatin Accessibility Profiling of Acute Myeloid Leukemia Reveals Heterogeneous Lineage Composition at Time of Therapy-Resistance

We very much appreciate reviewers' careful and thorough reading of our manuscript and their thoughtful comments, which have helped strengthen and clarify our work. Below, for your convenience, we quote the reviewers' original comments in *"italic blue"*, respond to each point in regular font, and include the most relevant revised text here in *"quoted red"*. Manuscript was revised in response to reviewers' comments with tracked changes.

"In this study, the authors investigated the heterogeneous lineage composition of AML cells at time of therapy resistance using single-cell ATAC-seq assay. They found that therapy resistant AML cells were primed for stem and progenitor lineages as well as various differentiated lineages. They further claimed the lineage composition revealed by the chromatin accessibility features could be used for prognosis prediction in TCGA AML cohort."

In general, this study provides a new view of lineage composition of therapy resistant AML cells by profiling the chromatin accessibility in single cells comparing with previous reports by single cell transcriptomes. These data could be a nice resource and of great interest to researchers in AML field. The analysis in this study is quite clear and solid."

We much appreciate your positive comments on our manuscript, and valuable suggestions which have helped enhanced our manuscript significantly.

"1, What are the major difference in term of chromatin accessibility between each AML malignant lineages and their normal counterpart?"

Thank you for your comment. We observed that chromatin accessibility tended to open up when comparing malignant lineages with their normal counterparts (Figure 3A; Figure S6).

Added Results.

*"To further characterize the diverse AML cell states, we integrated scATAC-seq and scRNA-seq data to link cell states-associated DARs with DEGs **by comparing malignant cell states with their normal counterparts** (Figure S6, Table S1). As shown, monocyte- and MEP-like AML cell states in scATAC-seq space resembled the monocyte- and early erythroid-like AML cell states in scRNA-seq space, respectively. In particular, open DARs **corresponded** with up-regulated DEGs in the same or similar AML cell states across both modalities. **Compared to monocyte- and MEP-like AML cell states, HSPC-like AML cells showed fewer differential events in chromatin accessibility. We also observed that chromatin accessibility tended to be more open when comparing malignant lineages with their normal counterparts (Figure S6).** Differential events were then binarized to retain information for differential directions (Figure 3A)."*

"2, Are there many repeat element regions being activated in therapy resistant AML cells?"

We thank the reviewer for this informative comment. Study regarding repeat element is an important perspective to look at, however, this is out of our current research scope and expertise. We will surely take it into consideration in our future studies.

REVIEWERS' COMMENTS:

Reviewer #1 (Remarks to the Author):

Fen, Wang and colleagues addressed most of the underlined points with additional analyses or further discussion in the text.

Major points in detail:

1) A clear schema has been included in Figure 1 to summarize the cohort and the overall study.

2, 10-13) As concerns the suggestion to further elaborate on the relationship between transcription and epigenetic regulation, authors acknowledge the only partial investigation of such interplay between layers in favor of a more focused profiling of chromatin accessibility (I agree, and it is well established in literature, this interplay is not expected to translate into high correlation between the two). Despite this, a clear definition of the integration strategy and other technical details on the analysis have now been reported in the Methods section and Results where appropriate.

Two further specifications:

point 11 - I would still believe the DAR call on assigned state vs all other assigned+unassigned cells (as done to identify cluster markers basically) would better fit.

point 13 - I would suggest to further clarify line 239 as follows "DAR-DEG pairs were kept only if both showing a significant differential regulation in the same direction"

3) A valuable investigation of the longitudinal hallmarks of patient disease evolution has now been added by authors, probably worth to be shortly addressed also in the Discussion.

5) An annotated UMAP in Figure 1 has been included as suggested, with label transfer robustness further confirmed with alternative algorithms. Would be probably helpful and transparent to plot such UMAP by key metadata to confirm any batch resolution.

6-9) All points were addressed exhaustively.

Overall, I believe the reported manuscript improvements make it suitable and ready for publication. I would only appreciate a further revision of points 3 (reference to longitudinal analysis in the Discussion) and 13 (line 239, sentence clarification on DEG-DAR pairs).

Reviewer #2 (Remarks to the Author):

Fan et al have made changes according to requirements. However they need to increase resolution of the supplementary Figure S2.

The data-set presented by Fan et al. would be very useful. According to current state of art their analysis is updated. I can not hold them responsible, however the analysis is still far from ideal. However it is Ok to publish it after minor corrections and proof reading

Reviewer #3 (Remarks to the Author):

My concerns have been addressed.

Response to the comments of Editors and Reviewers

Manuscript ID: COMMSBIO-23-0253A

Title: Single-Cell Chromatin Accessibility Profiling of Acute Myeloid Leukemia Reveals Heterogeneous Lineage Composition upon Therapy-Resistance

We appreciate editors' and reviewers' comments which further help improve the quality of our work. Particularly, we thank Reviewer 1's critical reading of our manuscript which makes us appreciate even more of the efforts and time devoted with serving as a peer reviewer.

Below, for your convenience, we quote the editors' and reviewers' original comments in *"italic blue"*, respond to each point in regular font, and include the most relevant revised text here in *"quoted red"*. At this stage, only minor editing to follow formatting guidelines is made to the final version of our revised manuscript.

"At the same time we ask that you edit your manuscript to comply with our format requirements and to maximise the accessibility and therefore the impact of your work.

** Please see the attached document for editorial requests for the final version (.docx file). Please ensure a completed version of this file is uploaded as a Related Manuscript with your final submission.*

** Please review our final submission file checklist to ensure all necessary files are present with your final submission and to avoid delays in accepting your manuscript. For your reference, a style and formatting guide is available here and includes all of our style requirements.*

** An updated editorial policy checklist that verifies compliance with all required editorial policies must be completed and uploaded with the revised manuscript. All points on the policy checklist must be addressed; if needed, please revise your manuscript in response to these points. Please note that this form is a dynamic 'smart pdf' and must therefore be downloaded and completed in Adobe Reader. <https://www.nature.com/documents/nr-editorial-policy-checklist.pdf>*

We have addressed the above editorial requests.

"Reviewer #1: Overall, I believe the reported manuscript improvements make it suitable and ready for publication. I would only appreciate a further revision of points 3 (reference to longitudinal analysis in the Discussion) and 13 (line 239, sentence clarification on DEG-DAR pairs)."

We appreciate reviewer 1's evaluation and efforts to clarify that our revision and responses are sufficient to be accepted. As you suggested, longitudinal analysis is now referenced in the Discussion. Line 239 is now clarified according to your wording.

Line 239 is now clarified as follows.

“DAR-DEG pairs were kept **only if both showing a significant differential regulation in the same direction**, for instance open DAR and up-regulated DEG, or closed DAR and down-regulated DEG.”

In our second revision, longitudinal analysis was referenced in the Discussion.

“Importantly, subsequent treatment did not deter the heterogenous lineage composition of AML cells. Rather, AML cells maintained their stem cell-like epigenetic landscape, while propagating the hierarchal diversity in the lineages, although the relative abundance of some of those lineages changed with time. **Our patient-derived longitudinally inferred trajectory with supervised pseudotime ordering revealed the transition of cellular states based on the dynamic changes of chromatin accessibility following treatment.** Since the cell of origin can impact the response to therapy 11, we propose that the acquisition of a differentiation spectrum not only contributes to the generation and maintenance of leukemic stem cells as previously suggested, but it may also create distinct barriers for therapeutic responses ultimately leading to resistance and persistence of AML.”

“Reviewer #2: Fan et al have made changes according to requirements. However they need to increase resolution of the supplementary Figure S2.”

Thank you for your further comments. Resolution of Figure S2 is now boosted.